# AN EEG DATASET OF WORD-LEVEL BRAIN RESPONSES FOR SEMANTIC TEXT RELEVANCE

## ABSTRACT

Electroencephalography (EEG) can enable non-invasive, real-time measurement of brain activity in response to human language processing. Previously released EEG datasets focus on brain signals measured either during completely natural reading or in full psycholinguistic experimental settings. Since reading is commonly performed when considering certain content as more semantically relevant than other, we release a novel dataset for semantic text relevance containing 23,270 time-locked ($\sim0.7s$) word-level EEG recordings acquired from participants who read both text that was semantically relevant and irrelevant to self-selected topics. Using these data, we present benchmark experiments with two evaluation protocols: cross-subject and within-subject on two prediction tasks (word relevance and sentence relevance). We report the performance of five well known models on these tasks. Our dataset and code are openly released. Altogether, our dataset paves the way for advancing research on language relevance and psycholinguistics, brain input and feedback-based recommendation and retrieval systems, and development of brain-computer interface (BCI) devices for online detection of language relevance.

## 1 INTRODUCTION

Human cognition is remarkably adept at attending to information that is specifically relevant to an individual's goals (Dwarakanath et al., 2023; Breton-Provencher et al., 2022; Bucher & Schumacher, 2006; Henderson et al., 2009). This ability of attending to salient information is also well known in research on language, which has repeatedly shown that content is facilitated in language processing if it matches individual interests, prior knowledge, and current goals (McCrudden & Schraw, 2009; Peng et al., 2018). Indeed, research in cognitive neuroscience has demonstrated that the human brain is capable of assessing whether text or even single words are relevant to a current information need within only a fraction of a second (Kotchoubey & Lang, 2001; Wenzel et al., 2017; Federmeier & Laszlo, 2009; Kutas & Federmeier, 2011). To this end, *relevance* of information has been extensively studied in the scope of information retrieval (Berger & Lafferty, 2017; Zhai et al., 2015), but most approaches have been based on signals captured from human behaviour and interactions, such as click-through data and dwell time (Joachims et al., 2005; 2017; Bi et al., 2020; MacAvaney et al., 2019) rather than directly from human cognition. Therefore, an intriguing alternative for behavioural signals is to infer relevance directly from the brain when a human is examining information. Previous seminal work has used brain signals to show that relevance responses reflect the graded importance of stimuli (Pinkosova et al., 2020). Predictive models using brain recordings have also been built to improve text word representation models (Hollenstein et al., 2021), and to estimate sentence relevance in question-answering (Gwizdka et al., 2017; Ye et al., 2022). However, these predictive models, and the datasets used, do not account for the significance of relevance when a human is reading text that holds more semantic relevance to them compared to other texts.

**The novelty of our work lies in the introduction of an original dataset that was recorded with the goal of capturing semantic text relevance through time-locked word presentation**, which has not been done previously. We release data from participants who read Wikipedia documents that were either semantically relevant or irrelevant to self-selected topics. Sentences from the documents were presented word by word on a screen for a fixed duration, which ascertained EEG recordings were minimally affected by auxiliary confounding factors, such as those related to participant's task engagement and eye movement patterns. Indeed, such confounding effects are well known to occur during naturalistic reading tasks in which users read whole sentences presented at once (Hollenstein

et al., 2020); for example, relevant words tend to be focused on for longer duration, thereby altering the extent to which EEG recordings include oculomotor activity – one of the strongest contributors to raw EEG recordings (Spapé et al., 2015; Spapé, 2021). Therefore, EEG recordings that are time-locked to single words with constant duration ensure that the relevance manifested in the brain responses corresponds to the exact word being read by the participant.

We present benchmark experiments using our dataset on two tasks: word relevance classification and sentence relevance classification. In the word relevance classification, the task is to estimate the semantic relevance of each word occurring in a Wikipedia document to the topic of the document. In the sentence classification, the task is to estimate the semantic relevance of each sentence in a Wikipedia document to a self-selected topic by a participant. We report the performance of five models on these tasks to allow researchers to compare and fairly assess the performance of machine learning models and their generalisation potential to unseen users.

## 2 NEUROPHYSIOLOGICAL DATASETS OF HUMAN LANGUAGE PROCESSING

In recent years, a variety of neurophysiological data collection procedures have been performed to record brain signals from participants performing reading tasks. However, only a limited amount of collected datasets have been released. Table 1 provides an overview of the neurophysiological datasets of human language processing that are publicly available. All except four of the listed datasets are based on EEG to acquire participant's brain recordings. This can be explained by the portability, costs, and practicality of EEG devices for real-world applications compared to fMRI and MEG, which have a higher spatial resolution and allow studying the regions of the brain characteristic for language processing. However, these methods are restricted to laboratory studies and cannot be realistically used for human-computer interaction, such as brain-computer interfacing. The datasets also vary in terms of the stimulus modality perceived by the participants: listening (auditory) and reading (visual). During listening tasks, a participant listens to a recorded utterance while brain responses are acquired. The collected brain responses require knowing the word boundaries to extract brain responses for each word (Schoffelen et al., 2019; Broderick et al., 2019). During the recording of brain responses of humans performing reading tasks, a single word or a sequence of words (i.e. sentence) is presented at once. Although the presentation of a whole sentence represents a more natural reading scenario, the correspondence of brain recording to single words is not possible a priori and requires some auxiliary information. For example, Hollenstein et al. (2018; 2020) used eye tracking data to time-lock EEG recording with eye fixations on words. However, eye fixations may limit the data (ERP window captured) as well as increase the chance of eye movement artefacts in EEG. In some cases, even when a word is defined as a stimulus event, the recording of brain responses may span several words (Wehbe et al., 2014). Thus, datasets that use time-locked stimulus recording provide a more reliable and distinguishable signal that is not influenced by external artefacts, e.g., eye movements. Although reading tasks that require participants to read words or sentences without any particular objective are helpful for understanding human language processing, they cannot be applied to application scenarios that require participants to perform a task with a specific goal in mind. The dataset closest to ours is collected by Ye et al. (2022), which contains brain responses acquired in a question-answering task. However, the objective of their neurophysiological data acquisition procedure is different from ours. We let the participant freely select the topic and keep in mind the selected topic during a reading task, thus imitating a natural process of reading. In their EEG data collection procedure, participants read sentences as answers to questions that could be perfectly relevant, relevant, or irrelevant, thus creating a more artificial scenario.

## 3 EEG DATA ACQUISITION

### 3.1 PARTICIPANTS

Volunteers were recruited via convenience sampling and by advertisement on university mailing lists targeting student populations. Following initial interest, online measures of handedness (Edinburgh Handedness Inventory) and English fluency (Cambridge English Adult Learners fluency test) were taken. The participants demonstrated high fluency, with an average score of 23.53 (SD = 1.23) on a standardised English proficiency test Cambridge University Press (2024), where the maximum possible score is 25. This score reflects strong English language skills, supporting the claim of "high

Table 1: Comparison of publicly available neurophysiological datasets of human language processing. We consider only the datasets that contain brain responses acquired from participants in response to visual reading of text or listening to spoken language. s: sentence. w: word. v: visual, a: auditory. u: utterance. - the value is not provided in the original publication. ⋆ the elapsed amount of time between consecutive recorded brain volumes. † word boundaries were extracted by the means of eye fixations. * brain responses acquired in a question-answering task rather than just reading.

| Dataset | Modality | # Participants | Stimulus type | Stimulus event | Time-locked event recording | Specific task | # Total stimulus events | # Recordings |
|---|---|---|---|---|---|---|---|---|
| Wehbe et al. (2014) | fMRI | 9 | v | w | no | no | ~5,000 | 11,250⋆ |
| Schoffelen et al. (2019) | MEG | 102 | a | u | no | no | - | - |
| Schoffelen et al. (2019) | fMRI | 102 | v | w | no | no | - | - |
| Nastase et al. (2021) | fMRI | 345 | a | u | no | no | 27 | 369,496⋆ |
| Hollenstein et al. (2018) | EEG | 12 | v | s | no | yes | 407 | 8,164† |
| Hollenstein et al. (2020) | EEG | 18 | v | s | no | yes | 390 | 8,310† |
| Broderick et al. (2019) | EEG | 19 | a | u | no | no | 1 | - |
| Broderick et al. (2019) | EEG | 19 | v | w | yes | no | - | - |
| Ye et al. (2022) | EEG | 21 | v | w | yes | yes* | - | - |
| Murphy et al. (2022) | EEG | 1 | v | w | yes | no | 404,205 | 404,205 |
| Our dataset | EEG | 15 | v | w | yes | yes | 23,270 | 23,270 |

English fluency". We believe that the used test provides a reliable indication of participants' reading proficiency. Participants were selected only if they were right-handed, had high English fluency, and were of good mental health (self-reported). Seventeen participants conformed to these criteria and participated in the study, which was conducted in line with the principles of the ANONYMOUS. Conforming to the standards laid out in the ANONYMOUS, participants received full instruction on the study's nature and objectives, and were informed on their rights as participants, including the right to withdraw from the study at any time without fear of any consequences. Two film tickets were given in compensation for their time (up to two hours, including setup time) and effort. Data from two participants were discarded due to a technical error, and therefore the present data contains recordings from fifteen participants (eight female, seven male).

## 3.2 STIMULI

Stimuli were obtained by searching the English Wikipedia[1]. A convenience sampling of queries was carried out, with the selection criteria being that topics should be of common interest and that the returned result should provide a sufficiently descriptive summary of the topic within the first six sentences of the article. Only the first six sentences of each article were retained. All punctuation and non-textual information was scrubbed. After manual inspection of the results, 30 topics were retained: `cat`, `painting`, `atom`, `society`, `wife`, `wine`, `rome`, `star`, `school`, `brain`, `savanna`, `volcano`, `politics`, `schizophrenia`, `plato`, `communism`, `michael jackson`, `learning`, `bank`, `machine learning`, `bicycle`, `automobile`, `bill clinton`, `india`, `money`, `euro`, `time`, `ocean`, `telephone`, `football`. During brain data recording, all words were displayed in Lucida Console typefact (18 pt), presented word by word in the centre of the screen in black colour against a grey (RGB D2, D2, D2) background.

## 3.3 PROCEDURE AND DESIGN

Following the setup of the electrophysiological apparatus, providing instruction to the participants, and acquiring the signed informed consent from the participants, the data recording began.

---

[1]Wikimedia commons, source dump July 2014

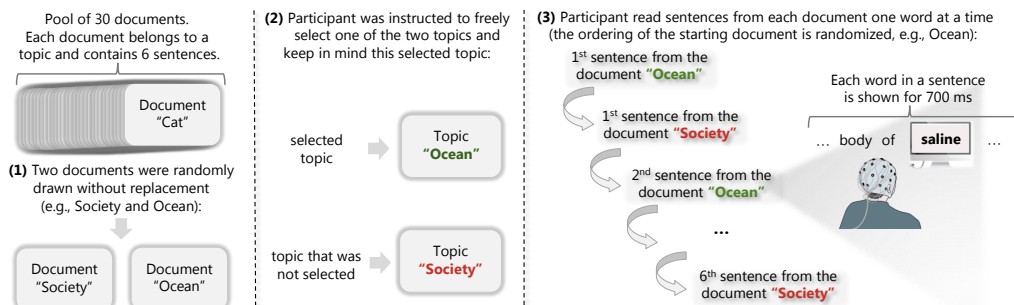

Figure 1: Step-by-step acquisition procedure of neurophysiological data during a reading task. (1) Each reading task contains two documents. Each document belongs to a particular topic. (2) A participant was asked to select one of the two topics and keep in mind the selected topic during a reading task. (3) The participant read sentences from two documents in alternated order word by word, such that a sentence was read from one document, then from the other document. While the participant read a word presented on a screen, the brain responses for that word were recorded.

This involved participants undertaking a series of eight *reading tasks*. As illustrated in Figure 1, in each reading task, two six-sentence documents were randomly drawn without replacement from the stimulus sample pool, and participants were requested to freely select one topic, to be assigned as the relevant topic. The participant was then instructed to keep in mind the selected topic that they chose while reading the documents, with the suggestion that at the end of the reading task, they would be asked to explain something about the relevant topic. By giving such an instruction to the participant, we provide an informational, intrinsic goal and ensure that the participant's focus is directed towards the words that are semantically relevant to the chosen topic. In other words, we simulate a natural setting in which participants search (by reading) for the information that is goal-relevant to them. However, to obtain further measures of topical relevance, participants were also asked at the end of each reading task to indicate how much they knew about each of the two topics on a scale of 1 (nothing) to 9 (everything), and how interesting they found the topics (1: not interesting, 9: extremely interesting).

Within each reading task of six sentences from two topics, each *trial* involved the two sentences being presented one after the other using a version of the common psychophysiological rapid serial visual presentation paradigm, aimed at minimising artifactual confounds in the EEG signal related to, for example, eye-movements, order effects, or visual differences. Each word within a sentence was sequentially presented at a steady pace of $\sim 700$ ms. Stimuli were shown always at the centre of the screen (to avoid eye-movements), against a visual mask (4 rows of 21 + signs: for standardising the luminance difference between long and short words). Before and after each sentence, a "separator" consisting of a non-alphabetic character or an integer (from 4 to 9) repeated seven times was shown to indicate the beginning and ending of the current sentence, followed by the switch to the other topic or the end of the trial. When that happened, participants were requested to repeat the name of the topic they had originally indicated as relevant, so as to ascertain their retention of the previous selection. The topics for reading tasks were randomly drawn from the total stimulus pool of 30 documents, sentences were presented in sequential order, but with their ordering within pairs randomised, to control for selectively focusing on the beginning or end of a trial.

## 3.4 APPARATUS

A Brain Products QuickAmp USB was used to digitise EEG recordings from passive Ag/AgCl electrodes placed at the 32 relatively equidistant sites of the 10/10 system of Fp1, Fp2, F7, F3, Fz, F4, F8, FT9, FC5, FC1, FC2, FC6, FT10, T7, C3, Cz, C4, T8, TP9, CP5, CP1, CP2, CP6, TP10, P7, P3, Pz, P4, P8, O1, Iz, and O2. A ground electrode placed at Fpz was used for the initial reference, but data were digitised with the common average reference. The stimulus presentation used a standard desktop LCD screen running in 1680 x 1050 resolution @ 60 Hz. E-Prime 2 (Psychology Software Tools Inc.) running on a Windows PC was used to ensure the timing accuracy of the synchronisation between runtime, EEG signals, and stimulus presentation.

Table 2: Summary statistics of our dataset after the EEG preprocessing. The value inside parentheses denotes standard deviation.

| # Sentences | # Unique stimulus events | # Total stimulus events | Average length of a sentence | # Documents each participant read | # Stimulus events annotated as semantically relevant to a topic |
|---|---|---|---|---|---|
| 1,440 | 1,401 | 23,270 | 16.2(6.5) | 16 | 7,155 |

## 4 DATA PREPROCESSING, ANNOTATION, AND ANALYSIS

### 4.1 EEG PREPROCESSING

Standard preprocessing of EEG data included, first, filtering the EEG signal by applying 35 Hz low-pass and 0.25 Hz high-pass filters. Then, a time window, ranging from $-200$ ms to $1000$ ms, was used to create equally sliced epochs of the EEG signal. An epoch represents the EEG signal of one word. The data of each epoch were corrected by subtracting the mean of a baseline period $[-200, 0]$, where $0$ denotes the onset of a stimulus. Finally, the EEG signal was cleaned by standard removal of signal fluctuations caused by eye movements or extreme noise levels. The processing of EEG data was carried out using the MNE library (Gramfort et al., 2013). Table 2 provides summary statistics of our dataset after the EEG preprocessing has been applied. Of the 16 documents that each participant read, the 8 documents belong to the selected topic, and the other 8 do not belong to the selected topic.

### 4.2 GROUND TRUTH ANNOTATION FOR WORD-LEVEL SEMANTIC RELEVANCE

A separate relevance assessment was conducted at the word level by three external annotators. Wilm et al. (2021) have argued that three annotators are sufficient to have consistent performance and adding more annotators results only in minor performance improvements. As our task is relatively straightforward, three annotators represent a reasonable choice for our task. Annotators (1 female, 2 male) have an academic degree and are fluent in English. Annotators did not participate in the collection of EEG data. The task of annotators was to annotate each word as 1 (semantically relevant) or 0 (semantically irrelevant) with respect to the topic of a Wikipedia document. For example, `saline` and `water` are semantically relevant to the topic `Ocean` in a given document, while `contains` and `stated` are semantically irrelevant to the topic `Ocean` in that given document. Detailed annotation guidelines can be found in Section H of the Appendix.

The inter-annotator agreement was measured using Fleiss' Kappa = 0.69, indicating a substantial agreement between the annotators (Fleiss, 1971). A majority voting of the three annotators' assessments was used to mark the final label of each word. On average, $\sim31\%$ of the words per topic are semantically relevant, with a standard deviation of $\sim7\%$.

### 4.3 ERP ANALYSIS

To validate the dataset in relation to the psychophysiological literature and more precisely describe the effect of relevance on the Event-Related Potential (ERP), we extracted the averaged time-locked activity from 250-350 ms, 350-450 ms, and 500-700 ms. The averaged potential over these bins – roughly corresponding to P300, N400, and P600, and chosen based on the literature and the course of the global field power shown in Figure 2 – was extracted for F3, Fz, F4, C3, Cz, C4, and P3, Pz, and P4 for further analysis.

A four-factor repeated measures ANOVA was then conducted with *time* (300 vs 400 vs 600), *relevance* (relevant vs irrelevant), coronal *position* (frontal, centre, parietal), and lateral *position* (left, medial, right), as factors. To briefly summarise its outcome, we report it here only with regard to the significant effects of relevance. This was observed as the main effect, $F(1, 14) = 72.83$, $p < .001$, with generally relevant words evoking a more positive potential ($0.61 \pm 0.11\mu$v) than irrelevant words ($0.12 \pm 0.08\mu$v). Relevance also had a two-way interaction with lateral position, $F(2, 28) = 3.77$, $p = .04$, a three-way with coronal position and time, $F(2, 28) = 12.75$, $p < .001$, and was part of the four-way interaction, $F(4, 56) = 6.69$, $p < .001$. As this indicates the effect of relevance was

modulated by time and space (i.e., an interaction between relevance, positioning of electrodes, and time was observed), we inferred three latent ERP potentials and conducted a three-way repeated measures analysis for the three bins. Here, an interaction between relevance and time either occurs if an effect 1) is observed stronger at one time point than another; 2) changes in direction (positive in one time point, negative in another); 3) is present in one, but not another (e.g., present at 300 ms but not at 400 ms). The analysis for the 250-350 ms time window showed an interaction between relevance and both coronal position, $F(2, 28) = 6.40$, $p = .005$, and lateral position, $F(2, 28) = 5.03$, $p = .014$. Relevance affected frontal and central sites more than parietal, and left and medial sites more than right. In contrast, in the 350-450 bin, the effect of relevance was less localised, with only the coronal position interacting with relevance, $F(2, 28) = 4.16$, $p = .026$, the effect of which was distributed more towards central and parietal sites. Likewise for the final bin between 500 and 700 ms, the effect of relevance was relatively consistent, although more present at the left and central sites than at the right sites, resulting in a significant effect interaction between relevance and lateral position, $F(2, 28) = 3.61$, $p = .04$. Note that the modest level of significance in interactions is in stark contrast with a robust main effect of relevance across bins: $F(1, 14) = 110.13$, $31.75$, and $85.07$ for the 300, 400, and 600 ms bins respectively, all $p$s $< .001$.

The above ERP results are consistent with previous research showing effects on P3 and P6 (Eugster et al., 2016; Potts, 2004).

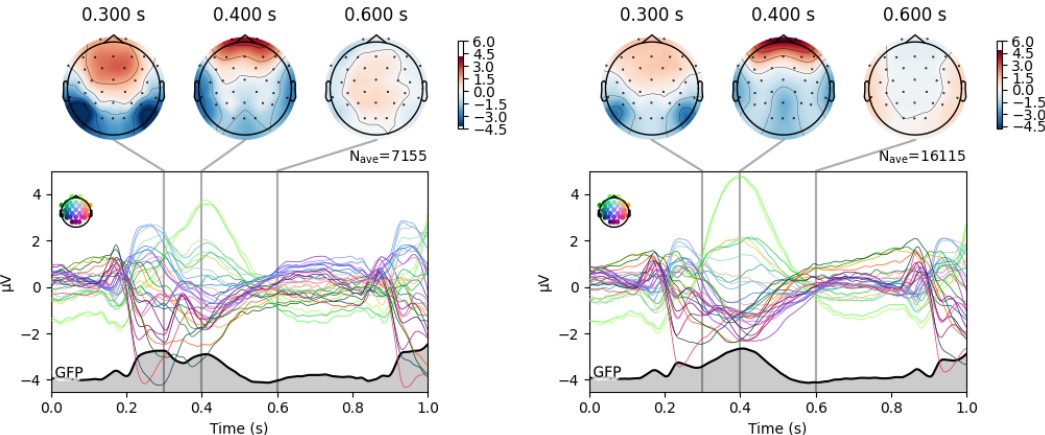

(a) Words annotated as semantically relevant to the topic of a document.

(b) Words annotated as semantically irrelevant to the topic of a document.

Figure 2: Evoked brain responses averaged across all words and participants for the words that are annotated as semantically relevant (a) and semantically irrelevant (b) to the topic of a document. The y-axis shows brain responses in microvoltages for each electrode (lines of different colours). The x-axis shows time progression of brain responses. The total number of semantically relevant and semantically irrelevant words used to calculate average responses are 7,155 and 16,115, respectively. The area at the bottom of two plots depicts the global field power (GFP) calculated as a spatial standard deviation over the brain responses. The comparison of ERPs between semantically relevant and semantically irrelevant words for each electrode are presented in the Figures 5, 6, 7, and 8.

## 5   BENCHMARK MACHINE LEARNING EXPERIMENTS

We benchmark our dataset in two classification tasks: word relevance prediction (Sec. 5.4) and sentence relevance prediction (Sec. 5.5). Each classification experiment is repeated 10 times, with a new random seed set for each run. Before that, we describe the training paradigms (Sec. 5.1), the classifiers (Sec. 5.2), and how the input EEG data are represented (Sec. 5.3).

## 5.1 Training paradigms

We adapt training and evaluation strategies similar to Huang et al. (2021); Zhang et al. (2022); Ding et al. (2022). Specifically, two paradigms are used to train and evaluate classifier models: *cross-subject* and *within-subject*.

**Cross-subject.** The cross-subject training paradigm is a $k$-fold cross-validation, where the dataset is split into $k$ consecutive folds and each fold contains the data belonging to one participant. During training, each fold is used then once as a test set and the $k-1$ remaining folds are used to create training and validation sets. The validation set contains data of a single participant from a randomly selected fold. The training set contains the remaining $k-2$ folds. Since each participant performed eight reading tasks, we split the test set into eight test sub-sets, each containing the two documents from a single reading task. We evaluate the models on those test sub-sets. Thus, for $p$ participants, we have $p$ different training sets, $p$ different validation sets, $p \cdot 8$ different test sub-sets.

**Within-subject.** Participant-specific models are created by fine-tuning the models trained with a *cross-subject* paradigm on a participant's data belonging to a test set. For this, we perform an 8-fold cross validation. At each iteration of cross-validation, the training set contains data of six reading tasks, the validation and test sets contain data of one reading task each. The models trained with a cross-subject or within-subject strategy use exactly the same test set.

## 5.2 Classifiers

Five different classification models are used: a convolutional neural network (EEGNet), a linear discriminant analysis (LDA), a logistic regression (LR), a unified framework for EEG-based reading comprehension modelling (UERCM), and a recurrent neural network (LSTM). The EEGNet architecture (Lawhern et al., 2018) is widely used and consists of temporal and depthwise convolution operations for learning frequency and spatial filters, respectively. The LDA and LR models are widely used non-gradient methods to work with brain recordings (Blankertz et al., 2011; Davis et al., 2020; Banville et al., 2021; Ruotsalo et al., 2023). The UERCM model is an attention-based model that captures local interactions of EEG recordings within an input sequence (Ye et al., 2022). The LSTM architecture remains a popular choice for working with time series data such as EEG (Ren & Xiong, 2021; Freer & Yang, 2020; Zhang et al., 2021). For all gradient-based methods, we employ an early-stopping strategy while training the models. This means that we stop the training procedure when the performance on the validation set does not improve after one iteration on the whole training set. All gradient-based models are trained with a learning rate of 0.001, batch size of 30, binary cross-entropy loss, an Adam optimiser, and for at most 100 epochs. We use existing implementations of the models: EEGNet (Zhang et al., 2024), LSTM (Paszke et al., 2019), UERCM (Ye et al., 2022), LDA and LR (Pedregosa et al., 2011). We use the default parameters for these models if not otherwise stated.

## 5.3 EEG data representation

One of the major challenges of EEG signals and their application in machine learning is a relatively low signal-to-noise ratio (Goldenholz et al., 2009; Zhu et al., 2019; Bricker, 2020). Thus, the selection of discriminative features is important. We use the approach described in Blankertz et al. (2011) to extract spatio-temporal features. The presentation of a word is limited to 0.7 seconds. However, in our benchmark machine learning experiments, we consider EEG recordings within the 250-950 ms range relative to the stimulus onset. The selection of this time range is based on neurolinguistic research showing that ERPs occurring 250 to 700 ms after stimulus perception are likely indicators of the relevance of language stimuli (Kim & Osterhout, 2005), meaning that recordings prior to 250 ms are insignificant for the present stimuli. We discard the $[0, 250]$ ms range also due to visual cues that are commonly present before 250 ms after stimuli onset (Knowland et al., 2013; Gutierrez-Sigut et al., 2022). We specifically did not want the visual potentials to affect the results. For example, we would expect different responses within the $[0, 250]$ ms range based on word length, as more photons (light) on the screen would evoke elevated potentials during this temporal segment. However, this is not a factor that we want to measure, as word length is not a factor we want to account for. Instead, we wanted to make sure that our ERP effects only account for relevance and semantic processing

of stimuli (independent of how much light on the screen their presentation requires). Therefore, on purpose, the $[0, 250]$ ms range data were ignored.

Specifically, we extracted the EEG signal within the 250 - 950 ms range from each epoch. Subsequently, we introduce two ways to represent EEG data as input to classification models: *matrix-* and *vector-based*. To create a matrix-based representation, we split the extracted EEG signal of one word $i$ into $s$ sections and calculated an average for each section and for each electrode $k$: $\boldsymbol{R}_i^{k \times s}$. The vector-based representation is created as follows: $\boldsymbol{x}_i = \tau(\boldsymbol{R}_i^{k \times s})$, where $\tau$ is a flatten operation that collapses $\boldsymbol{R}_i^{k \times s}$ into a vector $\boldsymbol{x}_i$ so that the data from all electrodes are appended one after another for each $s$. We used all available electrodes in all our benchmark experiments ($k = 32$). The value of $s$ is 151 for the EEGNet model, which is a default parameter (chunk_size) in the implementation of the EEGNet model (Zhang et al., 2024). The value of $s$ is 7 for the LDA, LR, LSTM, and UERCM models, which is selected based on acquiring equally sliced fragments of the EEG signal of 0.1 second each.

## 5.4 WORD RELEVANCE CLASSIFICATION TASK

**Goal and implementation details.** The evoked brain responses differ between the words that are annotated as semantically relevant and semantically irrelevant to the topic of a document, as visualized in Figure 2 and statistically analysed in Sec. 4.3. Therefore, our first benchmark is a classification problem, where we train models to predict if a word is semantically *relevant* or *irrelevant* to the topic of a document. Whether a word is semantically relevant or not is defined by the ground truth. For the EEGNet, LSTM, and UERCM models, we use a matrix-based representation, while for the LDA and LR models, a vector-based representation of EEG data as input. While for the EEGNet model the $2D$ input can be processed directly by the model, for the LSTM and UERCM models the $s$ in $\boldsymbol{R}^{k \times s}$ becomes the sequence length.

**Results.** Table 3 shows the performance of the five models on the word relevance classification task. The EEGNet, LSTM, and UERCM models, trained following a within-subject paradigm, achieve higher classification scores when compared to the corresponding models trained following a cross-subject paradigm. This is due to the fine-tuning on the data of a specific participant. In the Appendix (Section E.6) we discuss why the LDA and LR models, which are trained from scratch (these models do not support fine-tuning in the conventional sense), show lower performance in the within-subject paradigm compared to the cross-subject paradigm. We achieve state-of-the-art results (within-subject) when compared to the previously reported results (Eugster et al., 2014; 2016).

## 5.5 SENTENCE RELEVANCE CLASSIFICATION TASK

**Goal and implementation details.** We model semantic relevance at the sentence level. We chose sentences instead of documents due to the design of EEG data acquisition, where the presentation of sentences from two documents was alternated during each reading trial to avoid ordering effects. Whether a sentence is defined as semantically relevant or not depends solely on the choice of the topic selected by a participant. This means that if a sentence belongs to a document, whose topic was selected by a participant, it is defined as semantically relevant, otherwise not. For the EEGNet model, we use a matrix-based representation, while for the LSTM, UERCM, LDA, and LR models, a vector-based representation of EEG data as input. Since the EEGNet, LDA, and LR models cannot process time-series data a priori, we average EEG representations across all words in a sentence. For the UERCM and LSTM models, the number of words in a sentence defines the sequence length.

**Results.** Table 3 shows the performance of the five models on the sentence relevance classification task. Overall, the LSTM model achieves the best classification accuracy results with respect to the reported $AUC$ scores across the two training paradigms. As expected, gradient-based models show higher scores when these models are initially trained on the data of other participants and then fine-tuned on the data of the specific participant, while the LDA and LR models are trained only on the data of that specific participant. While previous studies have not used EEG data to predict sentence-level semantic relevance, related tasks that involve brain recordings for relevance estimation in Information Retrieval have been explored (Gwizdka et al., 2017; Ye et al., 2022).

Table 3: Word relevance and sentence relevance binary classification results averaged over all participants. The best scores are highlighted in bold. The value inside parentheses denotes standard deviation. * means that the model is trained from scratch in the within-subject paradigm, as fine-tuning is not supported for this model.

| Model | Cross-subject | | | Within-subject | | |
| --- | --- | --- | --- | --- | --- | --- |
| | AUC | Precision | Recall | AUC | Precision | Recall |
| | Word relevance classification task | | | | | |
| EEGNet | 0.64 (0.04) | 0.53 (0.10) | 0.08 (0.05) | 0.70 (0.03) | 0.57 (0.05) | 0.24 (0.09) |
| LDA* | **0.65** (0.04) | 0.53 (0.10) | 0.11 (0.06) | 0.63 (0.03) | 0.43 (0.04) | 0.37 (0.05) |
| LR* | 0.64 (0.04) | 0.51 (0.09) | **0.16** (0.05) | 0.63 (0.03) | 0.42 (0.04) | 0.39 (0.04) |
| LSTM | 0.64 (0.03) | **0.57** (0.27) | 0.03 (0.03) | **0.82** (0.03) | **0.71** (0.03) | **0.48** (0.06) |
| UERCM | 0.61 (0.03) | 0.56 (0.20) | 0.03 (0.03) | 0.70 (0.03) | 0.62 (0.04) | 0.21 (0.07) |
| | Sentence relevance classification task | | | | | |
| EEGNet | 0.55 (0.06) | 0.55 (0.17) | 0.24 (0.17) | 0.75 (0.08) | 0.68 (0.08) | 0.67 (0.14) |
| LDA* | 0.72 (0.07) | 0.68 (0.07) | 0.56 (0.10) | 0.54 (0.05) | 0.52 (0.04) | 0.54 (0.07) |
| LR* | 0.71 (0.06) | 0.68 (0.05) | **0.58** (0.09) | 0.54 (0.06) | 0.54 (0.05) | 0.56 (0.07) |
| LSTM | **0.79** (0.06) | **0.83** (0.30) | 0.14 (0.18) | **0.97** (0.02) | **0.94** (0.04) | **0.82** (0.09) |
| UERCM | 0.67 (0.08) | 0.69 (0.12) | 0.38 (0.16) | 0.92 (0.04) | 0.85 (0.06) | **0.82** (0.07) |

Compared to these methods, our results in the within-subject sentence relevance classification task show significantly better performance.

# 6 DISCUSSION

**Limitations and future work.** While the application of EEG devices to everyday human-computer interaction in real-world settings is still under development, our work represents a significant advancement in addressing this challenge. Our EEG data acquisition approach controls for ordering effects and is designed to minimize confounding factors, ensuring validity and balance for downstream experimentation. However, due to the enhanced experimental control, our data may not fully reflect naturalistic, real-world interactive use. For instance, a block design with simultaneous presentation of all words within a topic would likely increase *ecological validity*. However, avoiding block design would result in order effects and signal artefacts severely limiting the applicability of the dataset – an issue previously noted as problematic for previous data used for EEG experimentation with real-world stimuli (Li et al., 2021). Therefore, we believe that the present dataset will serve as a reliable benchmark for future research.

A sample size of 15 participants is consistent with the sample sizes used in comparable EEG studies, such as those by Ye et al. (2022) (21 participants, 465 sentences, approximately 4,600 words) and Hollenstein et al. (2018) (12 participants, 407 sentences, 8,164 words). Our dataset contains 23,270 time-locked EEG recordings, providing a substantial amount of data for model training and testing. Brysbaert (2019) demonstrates that, with well-designed experiments, even a small pool of participants can provide sufficient generalisability. Given that relevance is often subjective and may manifest in brain responses differently for different participants, we also think it is more important to have more samples per participant than data from more participants. Thus, we believe that our rigorously designed EEG data collection procedure, which is carefully controlled for ordering effects and confounding factors, ensures that the dataset is robust and generalisable for its intended tasks. Nevertheless, the participant pool is relatively homogeneous, and the collection of brain responses from under-represented groups and different cultural backgrounds could facilitate even better generalisation. We also cannot fully exclude the possibility of human error affecting the quality of the data, such as participants not fully understanding the tasks or experiencing fatigue during the collection of EEG data.

Finally, our machine learning experiments focused on providing benchmark results without auxiliary data or novel model architecture development. For example, the LDA and LR models do not consider the temporal structure of the data, in contrast to the EEGNet, LSTM, and UERCM models. Here, EEGNet is explicitly designed to capture spatio-temporal features. For the LSTM model, temporal dependencies are captured via hidden and cell states and the spatial features are embedded into the vectors of data that represent the recorded EEG responses for a specific time step. Similarly, the UERCM model captures temporal and spatial relationships via self-attention. As the temporal aspect was found to be one of the significant factors in ERP analysis, we anticipate that this dataset will be valuable in advancing future research on novel machine learning architectures that explicitly account for the temporal and spatial structure of the data (Zhang et al., 2022; Pan et al., 2024). Therefore, we encourage the research community to develop novel model architectures to surpass our benchmark results.

In addition, we encourage using our data to develop new wearable neuroimaging devices and associated signal decoding architectures. This can accelerate the development of new types of brain-computer interfaces that account for the relevance of information, thus enabling applications in assistive technologies, such as adaptive learning systems and personalised content delivery, based on user engagement and interest. Moreover, our benchmark experiments highlight the potential of machine learning models to decode semantic relevance from EEG data, offering a foundation for extending these capabilities to new application domains like brain-state driven entertainment, neurofeedback training (training memory and attention through brain-relevance feedback), and cognitive workload monitoring to optimise task assignment and performance.

**Ethical considerations.** Data were collected in accordance with the principles of the Declaration of Helsinki of the World Health Organisation. All participants were informed of their right to withdraw at any time without consequences and provided written consent, which included the agreement for their anonymised data to be published. We do not anticipate any negative societal impacts from the use of our proposed benchmarks. However, despite existing regulations on the handling of personally identifiable and sensitive information, neurophysiological data, such as EEG, presents unique challenges that remain only partially resolved. EEG data can potentially reveal private information, including personal opinions, feelings towards others, and emotional states. Caution is advised when using neuroimaging data for machine learning research and commercial applications, as these methods could be vulnerable to future misuse.

## 7 CONCLUSION

**We introduced the novel EEG dataset specifically designed to capture semantic text relevance through time-locked word presentation, which is not addressed by any currently available datasets.** We provide a detailed overview of other datasets of human language processing and how they compare to our dataset. Our benchmark experiments showed that semantic relevance can be successfully decoded on a word- and sentence-level. Our dataset enables studying novel downstream tasks and applications for (a) information retrieval (e.g., retrieving the documents that a user finds semantically relevant), (b) recommender systems (e.g., recommending information to a user that satisfies their information need), and (c) user engagement (i.e., understanding and predicting user interactions with the displayed language content).

### CODE AND DATA AVAILABILITY

The code allowing to reproduce data processing and experimentation will be publicly available at ANONYMOUS upon acceptance of the paper. Data are available at ANONYMOUS.

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

## A DATASET DOCUMENTATION

### A.1 LINKS TO DATASET, CODE, AND DOCUMENTATION

**Dataset repository.** The "raw" acquired EEG data, the preprocessed ("cleaned") EEG data for machine learning pipelines, as well as the ground truth annotation for word-level semantic relevance, can be accessed at the following URL: ANONYMOUS.

**Code.** Our code to reproduce the benchmark results can be accessed at the following URL: ANONYMOUS.

**Datasheet.** To ensure responsible use of our dataset, we provide a "Datasheet" that describes the intended use of our dataset, its contents, etc. We use the template suggested by Gebru et al. (2018). The datasheet can be accessed at the following URL: ANONYMOUS.

**Croissant metadata.** We make our dataset available at ANONYMOUS, which contains the metadata and the preprocessed ("cleaned") EEG data for the direct experimentation with our dataset.

**Dataset website.** Upon acceptance, we will create a separate website for the dataset, accessible at ANONYMOUS.

The links provided above will be stable and accessible.

## A.2 LICENCE AND RESPONSIBILITY STATEMENT

The dataset is released under a Apache Licence 2.0 licence. We state that we bear all responsibility for the content of the dataset in case of violation of rights and confirm the dataset licence. We confirm that the data released have been fully anonymised and have the explicit permission of the participants whose EEG data are released to be shared openly.

## A.3 MAINTENANCE AND CONTACTS

The authors of the paper that introduced the dataset are responsible for supporting, hosting, and maintaining the dataset.

Questions about the code: ANONYMOUS.

Questions about the EEG data collection procedure: ANONYMOUS.

Other inquiries: ANONYMOUS.

# B RAPID SERIAL VISUAL PRESENTATION (RSVP)

In our dataset, where each word is presented for approximately 0.7 seconds, we use principles similar to those underlying Rapid Serial Visual Presentation (RSVP) to ensure precise time-locked EEG recordings. RSVP is an effective method for collecting data in studies that require precise temporal alignment between stimuli and acquired data (Potter, 1984). A search for "rapid serial visual presentation" and "EEG" in Google Scholar for 2023 yielded 508 results, demonstrating its broad application in the field.

# C DETAILS ON EEG DATA ACQUISITION

## C.1 PLACEMENT OF ELECTRODES

The EEG recordings are acquired from electrodes placed at the 32 relatively equidistant sites of the 10/10 system: Fp1, Fp2, F7, F3, Fz, F4, F8, FT9, FC5, FC1, FC2, FC6, FT10, T7, C3, Cz, C4, T8, TP9, CP5, CP1, CP2, CP6, TP10, P7, P3, Pz, P4, P8, O1, Iz, and O2. Figure 3 shows the placement of the electrodes on the head of a participant during the EEG recording setup.

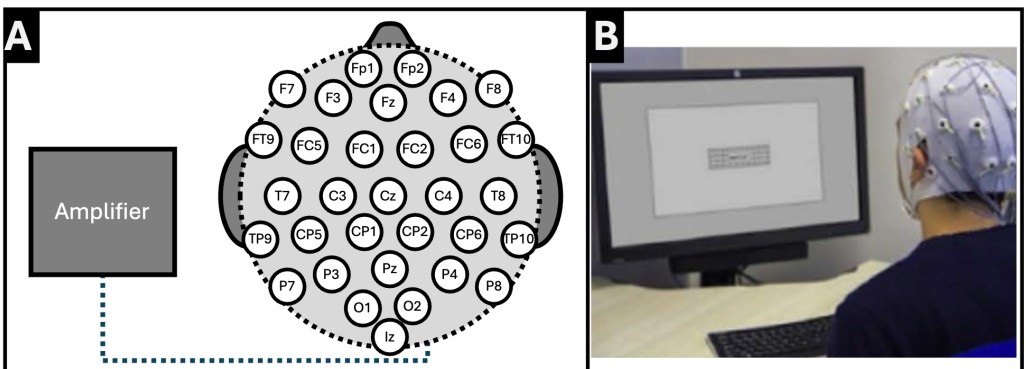

Figure 3: Placement of electrodes according to the 10-20 system (a) and EEG recording setup (b).

## C.2 STEP-BY-STEP ILLUSTRATION OF THE NEUROPHYSIOLOGICAL EXPERIMENT.

Each participant performed eight reading tasks in total. During each reading task, the participant read two documents, each belonging to a specific topic. Before each reading task, the participant had to select a topic from the two randomly drawn topics without replacement from the pool of 30 unique topics. Each reading task consisted of six reading trials. Each reading trial contained two

distinct sentences from two documents: one document belonging to a topic that was selected by the participant and another document belonging to a topic that was not selected by the participant. In each trial, a distinct sentence was shown from each document (the order of the sentences was preserved). Figure 4 illustrates the step-by-step neurophysiological experiment. for the acquisition of EEG data during a reading trial.

### C.3 DISCUSSION: BALANCING OF THE TOPICS

As each participant was presented with a unique pair of topics during each reading task and chose their preferred topic from pairs of topics randomly selected, exposure to specific topics varies among participants. However, complete balancing of the topics among the participants was not feasible due to the experimental design, which required the participants to choose one of the two topics they wanted to learn more about. This approach ensures that the participant's intrinsic interest or preference is prioritised rather than being influenced by enforced balancing. Figure 16 illustrates the frequency of topic selections among participants. Importantly, the unbalanced distribution of selected topics does not introduce inconsistencies in the dataset. For each reading task, one topic was selected and the other was not, establishing semantic relevance in an absolute manner. By "absolute", we mean that relevance is determined solely by the relationship between the selected topic and the unselected one, independent of other topics. Additionally, Figure 17 provides a breakdown of the number of words per topic. While some topics may have more words than others, we do not anticipate that this unbalanced distribution introduces inconsistencies in the dataset.

### C.4 ADDITIONAL ERP ANALYSIS

Additionally, to the ERP analysis presented in our paper, Figures 5, 6, 7,and 8 show the ERPs for the words that were annotated as semantically relevant and semantically irrelevant for each electrode.

To more clearly demonstrate the differences in ERPs between semantically relevant and semantically irrelevant words, we present in Figure 9 a topographic scalp plot showing the difference between ERPs for semantically relevant words and those for semantically irrelevant words. The topographic patterns of these differences align with the positivity patterns for the components P300, N400, and P600 reported by Eugster et al. (2016). Figure 10 illustrates the differences in ERP responses using the Fz, C3, C4, P3, Pz, and P4 electrodes. These differences are statistically significant ($p < 0.001$), as determined using a non-parametric bootstrapping method with 10,000 permutations.

### C.5 METADATA

Our dataset contains word-level brain recordings of participants reading Wikipedia documents. In the following, we refer to the metadata that are related to the preprocessed ("cleaned") EEG data and not the "raw" data, since the preprocessed data are the data that are intended to be used for machine learning tasks. However, other researchers are more than welcome to use the "raw" data for their needs. The "cleaned" data as well as the "raw" data are available at ANONYMOUS. Table 4 shows an example of metadata for one single instance (word-level EEG recording). A "cleaned" word-level EEG data contains 2001 EEG recordings for each electrode. These 2001 EEG recordings are voltage values and correspond to 1 second of a recording that starts from the stimulus onset (a word has appeared on a screen).

## D   SELF-REPORTED RATINGS OF INTERESTINGNESS AND PRE-KNOWLEDGE

Figure 11 shows self-reported ratings of interestingness and pre-knowledge of the topics for each participant.

## E   DETAILS ON CLASSIFICATION MODELS AND EVALUATION STRATEGY

We trained all our models without performing a hyperparameter optimisation and using in most cases the default parameters. This was intended as we wanted to provide the baseline benchmark results. If the default parameter was not used, we justify our selection of the value for that parameter.

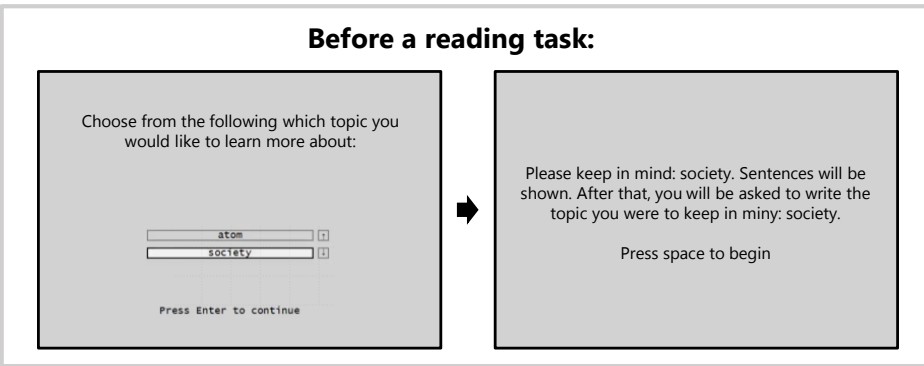

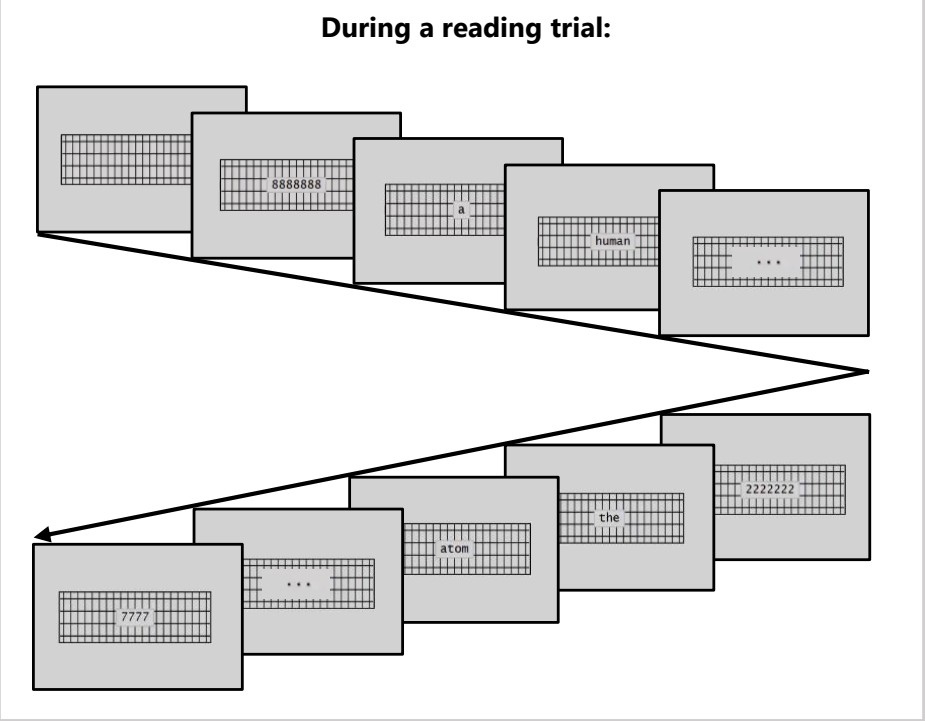

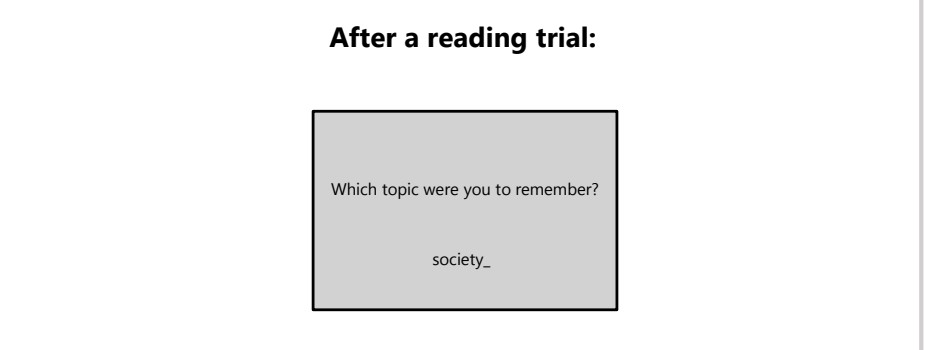

Figure 4: Step-by-step illustration of the neurophysiological experiment for the acquisition of EEG data.

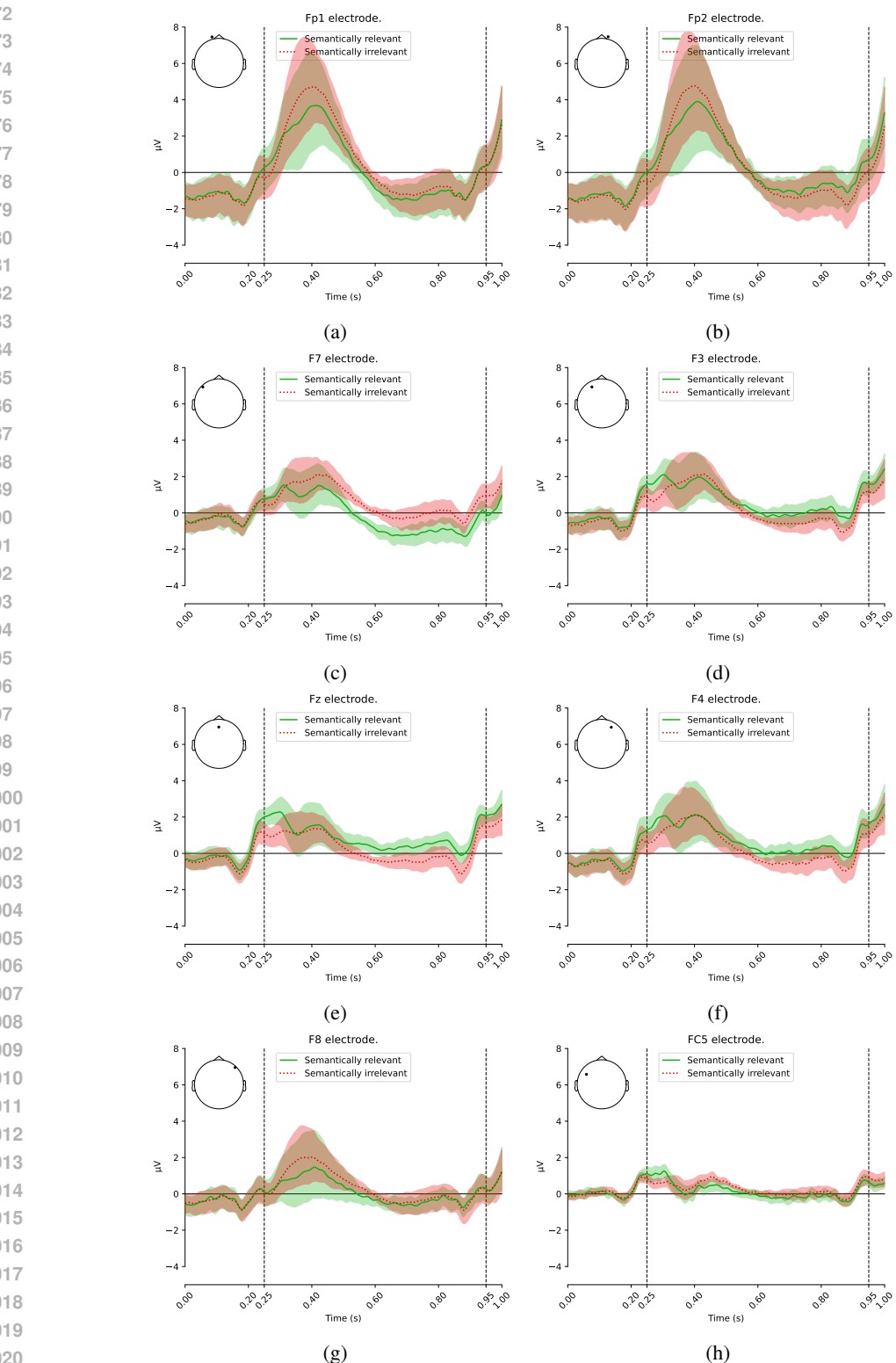

Figure 5: ERPs for the words that were annotated as semantically relevant and semantically irrelevant for the electrodes Fp1, Fp2, F7, F3, Fz, F4, F8, and FC5.

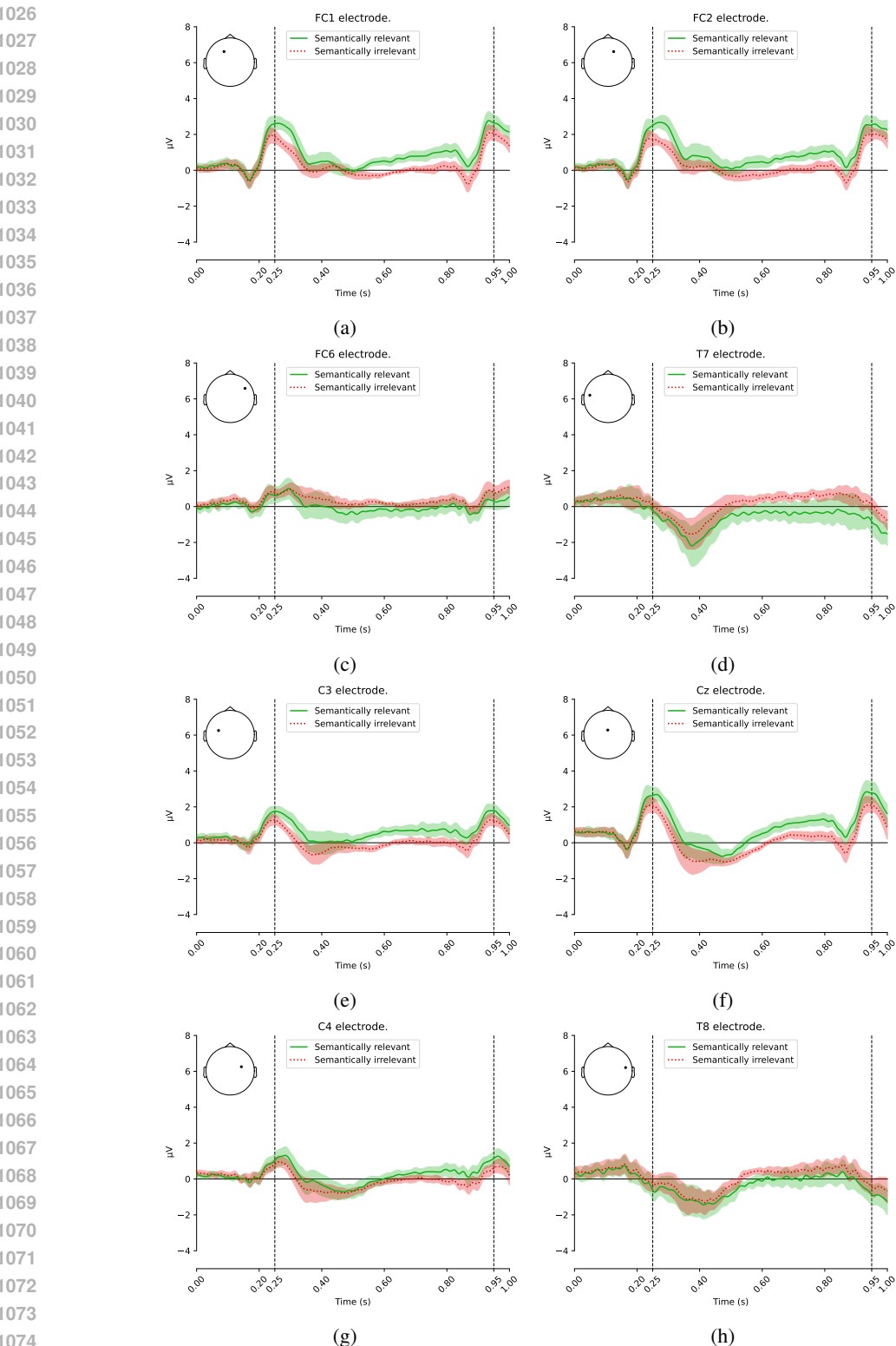

Figure 6: ERPs for the words that were annotated as semantically relevant and semantically irrelevant for the electrodes FC1, FC2, FC6, T7, C3, Cz, C4, and T8.

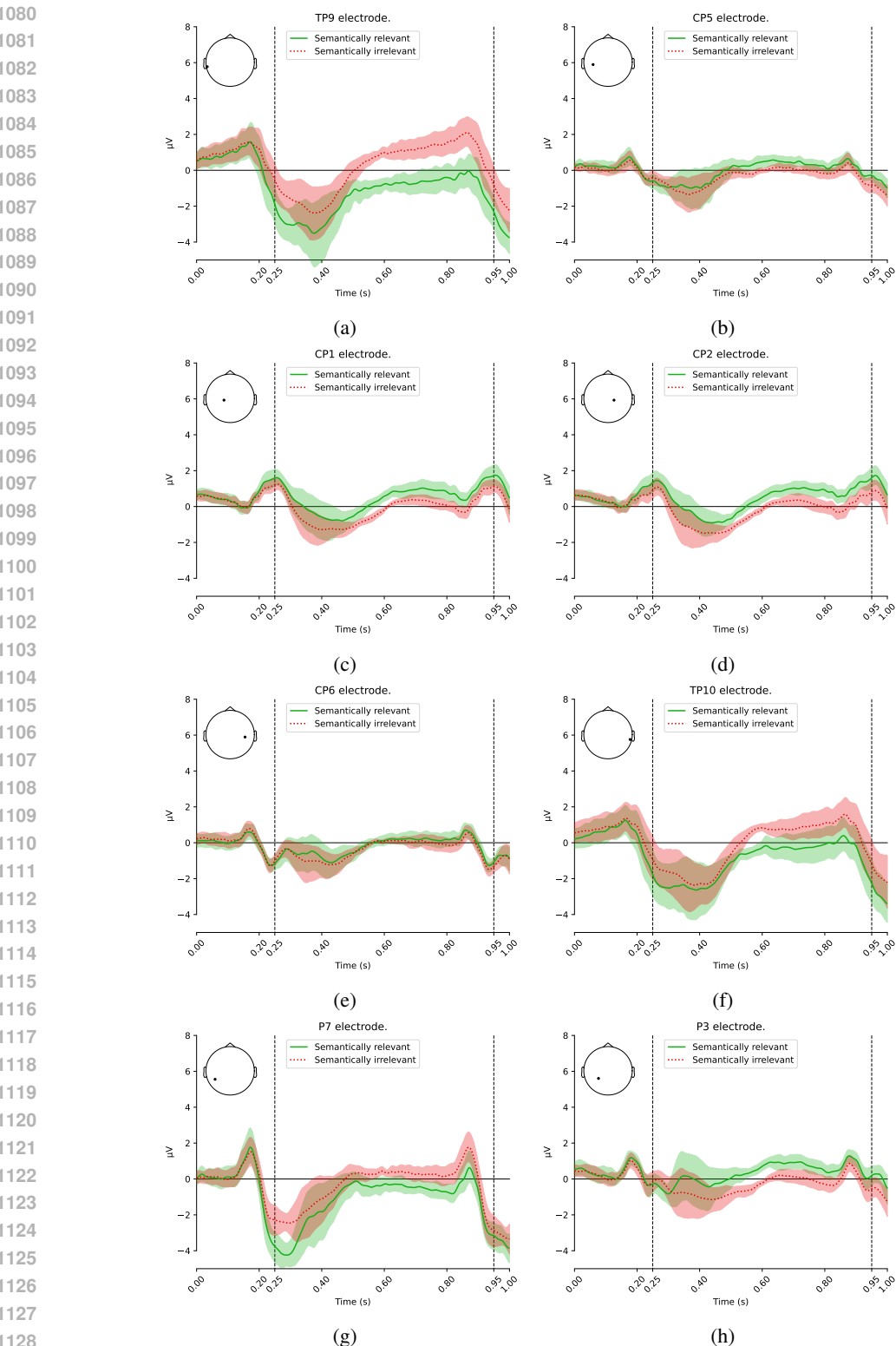

Figure 7: ERPs for the words that were annotated as semantically relevant and semantically irrelevant for the electrodes TP9, CP5, CP1, CP2, CP6, TP10, P7, and P3.

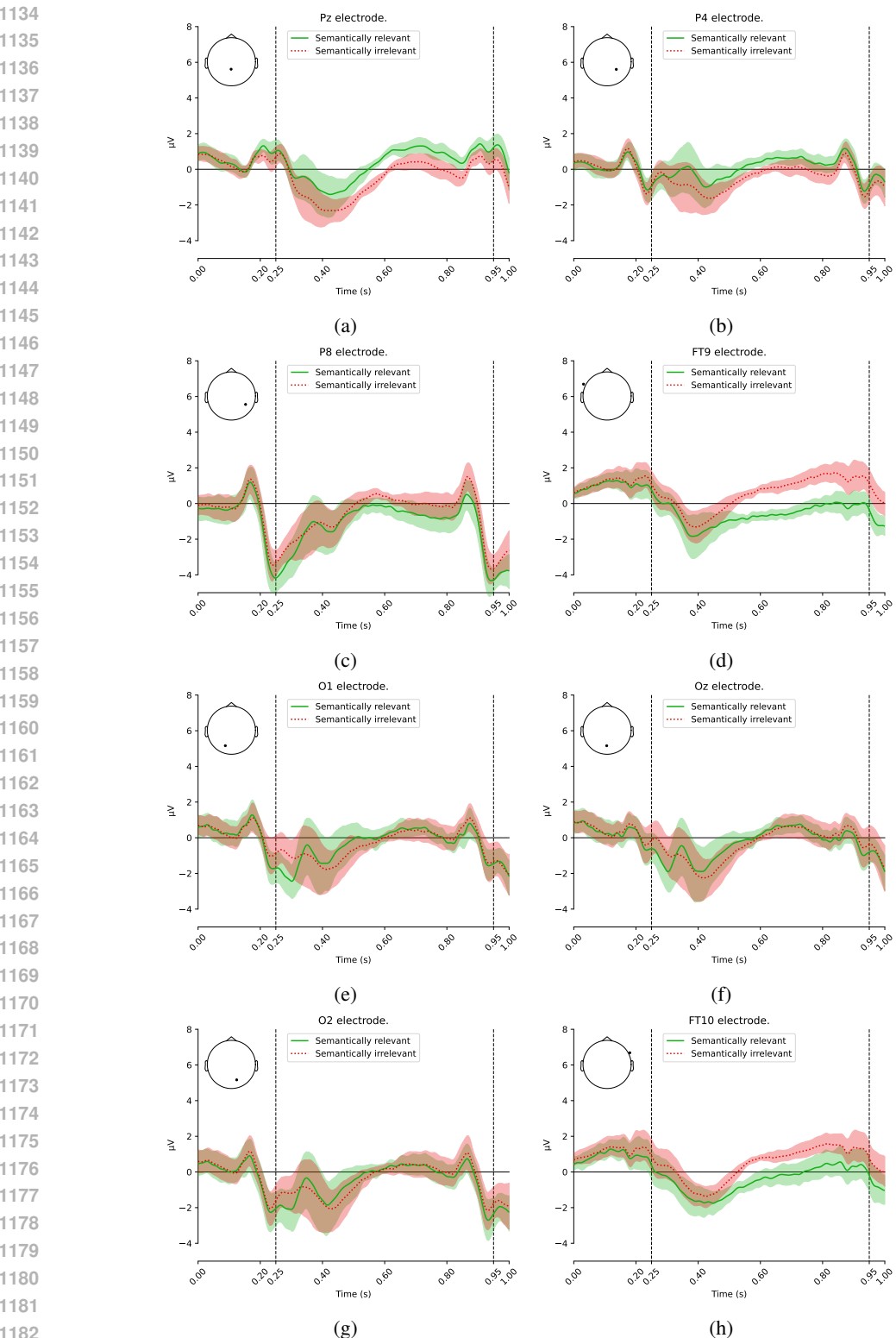

Figure 8: ERPs for the words that were annotated as semantically relevant and semantically irrelevant for the electrodes Pz, P4, P8, FT9, O1, Oz, O2, and FT10.

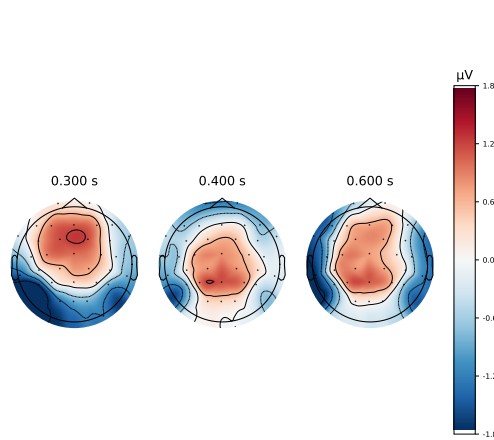

Figure 9: A topographic scalp plot showing the difference between the ERPs for semantically relevant words and those for semantically irrelevant words.

Figure 10: Population mean of ERPs obtained from the Pz, Fz, C3, C4, P3, and P4 electrodes. The differences are statistically significant (p < 0.001) using a non-parametric bootstrapping with 10000 permutations. Global field power (GFP) represents the standard deviation of the EEG signal across electrodes (visualized as shaded areas).

Table 4: An example of metadata pertained to each preprocessed ("cleaned") word-level brain recording instance. `Event` corresponds to a specific point in time during EEG data collection and represents the onset of an event (presentation of a word). `Word` is a word read by the participant. `Topic` is the topic of the document to which the `word` belongs to. `Selected topic` indicates the topic the participant has selected. `Semantic relevance` indicates whether the word is semantically relevant (expressed as 1) or semantically irrelevant (expressed as 0) to the topic selected by the participant. `Interestingness` indicates the participant's interest in the topic of a document. `Pre-knowledge` indicates the participant's previous knowledge about the topic of the document. `Sentence number` represents the sentence number to which the word belongs. `Participant` is the participant's ID.

| Event | Word | Topic | Selected topic | Semantic relevance | Interestingness | Pre-knowledge | Sentence number | Participant |
|-------|------|-------|----------------|--------------------|-----------------|---------------|-----------------|-------------|
| 13380340 | and | automobile | cat | 0 | 2 | 2 | 1 | TRPB101 |

## E.1 LOGISTIC REGRESSION

We have used the implementation of logistic regression provided by the scikit-learn library, version 1.4.2. The default parameters were used to train the model and were not changed in all benchmark experiments.

## E.2 LINEAR DISCRIMINANT ANALYSIS

We have used the implementation of linear discriminant analysis provided by the scikit-learn library, version 1.4.2). The default parameters were used to train the model and were not changed in all benchmark experiments.

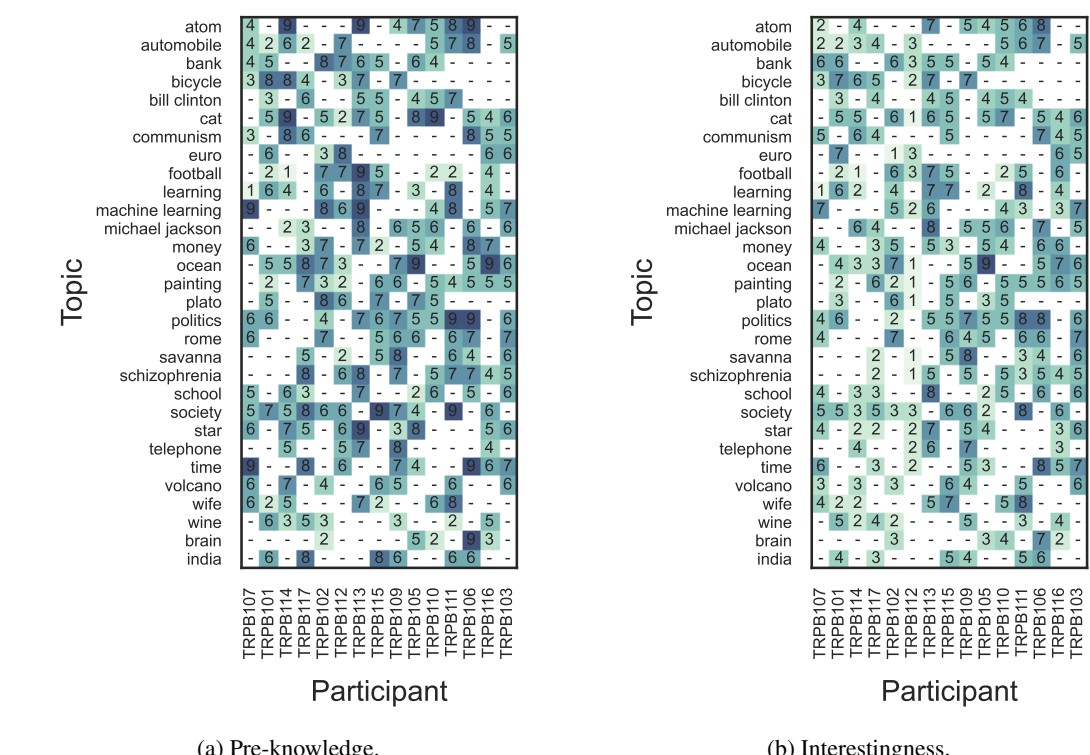

(a) Pre-knowledge.

(b) Interestingness.

Figure 11: Self-reported ratings of each participant's pre-knowledge (a) and interestingness (b) about each of the two topics presented during a reading task on a scale of 1 (nothing) to 9 (everything). - means that the topic was not presented to the participant for the selection as each participant read 16 documents. $X$-axis: participant ID. $Y$-axis: topic name.

### E.3 EEGNET

We have used the implementation of the EEGNet model provided by the torcheeg library, version 1.1.0). For all parameters, except `num_electrodes` and `num_classes`, the default values were used in all benchmark experiments. The parameter `num_electrodes` was set to 32 and represents all the electrodes available in our data. The parameter `num_classes` was set to 1, since we used binary cross-entropy loss.

### E.4 LSTM

We have used the implementation of the LSTM model provided by the PyTorch library, version 2.3.0. For all parameters, except `input_dim`, `hid_channels`, and `num_classes`, the default values were used in all benchmark experiments.

The parameter `input_dim` represents the dimensionality of the input EEG data. This parameter can be set to 224 or 32:

- The value of 224 is used when performing the *sentence relevance* classification task. The value of 224 corresponds to the vector representation of each word-level EEG data in a sentence, where the word-level EEG data are represented as a matrix with 32 rows and 7 columns. We have 32 rows, as we have 32 electrodes. Each row represents an electrode, and each column represents the averaged EEG signal within the 0.1s range over a time span of 0.25 to 0.95 s.

- The value of 32 is used when performing the *word relevance* classification task. The value of 32 corresponds to the number of electrodes.

The parameter `hid_channels` represents the number of features in the hidden state and was set to 32. We select 32 following simple reasoning: a single feature for each electrode.

The parameter `num_classes` was set to 1, since we used binary cross-entropy loss.

### E.5 UERCM

We have used the implementation of the UERCM model accessible at the following URL: `https://github.com/YeZiyi1998/UERCM/blob/master/UERCM/model.py`). For all parameters, except `feat_dim`, `max_len`, `d_model`, `num_layer`, and `num_classes`, the default values were used in all benchmark experiments.

The parameter `feat_dim` represents the dimensionality of the input EEG data. This parameter can be set to 224 or 32. The reason for selecting these values is the same as for the LSTM model.

The parameter `max_len` represents the input sequence length and can be set to 39 or 7:

- The value of 39 was used only during the `sentence relevance` classification task and corresponds to the longest sentence in all documents (i.e., the sentence that has the highest number of words). We select the value 39 to ensure that each sentence has the same length and can be put into a single `batch` containing many sentences. The sentences that have less than 39 words are padded with zeros. We ensure that padded data are not considered when training the model.

- The value of 7 was used only during the `word relevance` classification task and corresponds to 7 values produced by averaging EEG recordings for a single word over a time span of 0.25 to 0.95 seconds. Here, the first value represents the averaged EEG signal within the range of 0.25-0.35 s, the second value represents the averaged EEG signal within the range of 0.35-0.45 s, etc.

The parameter `d_model` represents the number of expected features in the encoder input and was set to 32. The reason for selecting 32 is the same as for the LSTM model setting the parameter `hid_channels` to 32.

The parameter `num_layer` was set to 2, as used by Pappagari et al. (2019) for document classification using a small Transformer architecture.

The parameter `num_classes` was set to 1, since we used binary cross-entropy loss.

### E.6 WHAT CAUSES THE LDA AND LR MODELS TO EXHIBIT INFERIOR PERFORMANCE IN THE WITHIN-SUBJECT PARADIGM WHEN COMPARED TO THE CROSS-SUBJECT PARADIGM?

In the within-subject paradigm, the LDA and LR models are trained from scratch, as these models do not support fine-tuning in the conventional sense. Consequently, they lack sufficient data in this paradigm to learn robust and generalisable features, resulting in lower performance compared to the within-subject paradigm, where a larger and more diverse dataset is available for training. This limitation is consistent with findings in the literature. For example, (Huang et al., 2021) also reported cases in which models trained on limited participant-specific data failed to outperform generic classifiers. Thus, the observed results are likely due to the restricted data availability in the within-subject paradigm.

## F    CLASSIFICATION RESULTS PER PARTICIPANT

Figure 12 and Figure 13 show binary classification results per participant for word and sentence classification tasks, respectively.

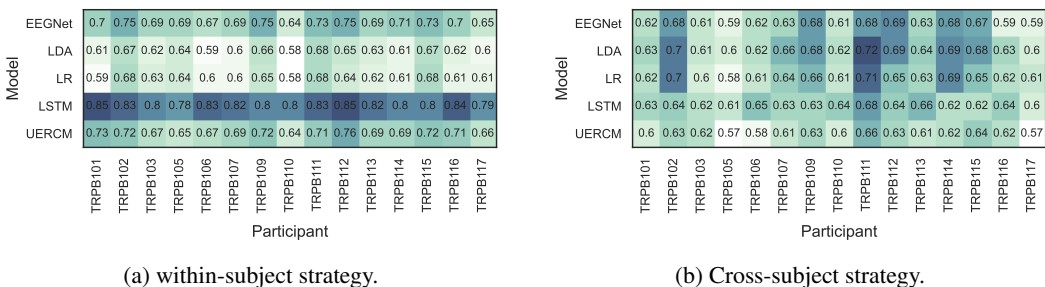

(a) within-subject strategy.                    (b) Cross-subject strategy.

Figure 12: Word relevance binary classification results per participant. A value inside each cell represents an averaged AUC score. $X$-axis: participant ID. $Y$-axis: model. Darker colour means higher AUC score.

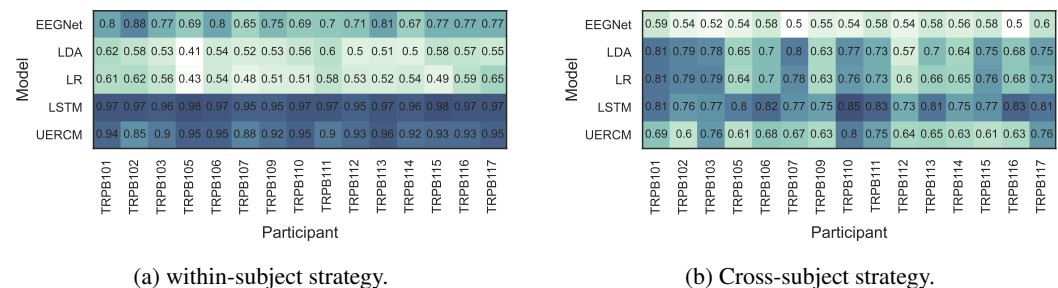

(a) within-subject strategy.                    (b) Cross-subject strategy.

Figure 13: Sentence relevance binary classification results per participant. A value inside each cell represents an averaged AUC score. $X$-axis: participant ID. $Y$-axis: model. Darker colour means higher AUC score.

## G    OVERLAP OF SEMANTICALLY RELEVANT WORDS ACROSS TOPICS

We analyse the overlap of semantically relevant words across topics. Since Wikipedia articles contain text on specific or specialised topics, we expect minimal overlap in semantically relevant words. Figure 14 shows that the overlap between all semantically relevant words across topics in our dataset is up to 25%.

## H    ANNOTATION GUIDELINES FOR SEMANTIC RELEVANCE

The goal of this annotation task is to determine whether each word in a document is semantically relevant or semantically irrelevant to the topic of the document. Only the intrinsic semantic relevance of each word alone to the topic must be assessed.

### H.1    KEY DEFINITIONS

- **Semantic Relevance:** A word is semantically relevant if it contributes meaningfully to the topic of the document by either directly describing or being closely associated with it.
  *Example:* For the topic "Ocean", words like *water*, *saline*, *wave*, and *marine* are relevant.

- **Semantic Irrelevance:** A word is semantically irrelevant if it does not provide meaningful information about the topic.
  *Example:* For the topic "Ocean", words like *a*, *influences*, *primary*, and *divided* are irrelevant.

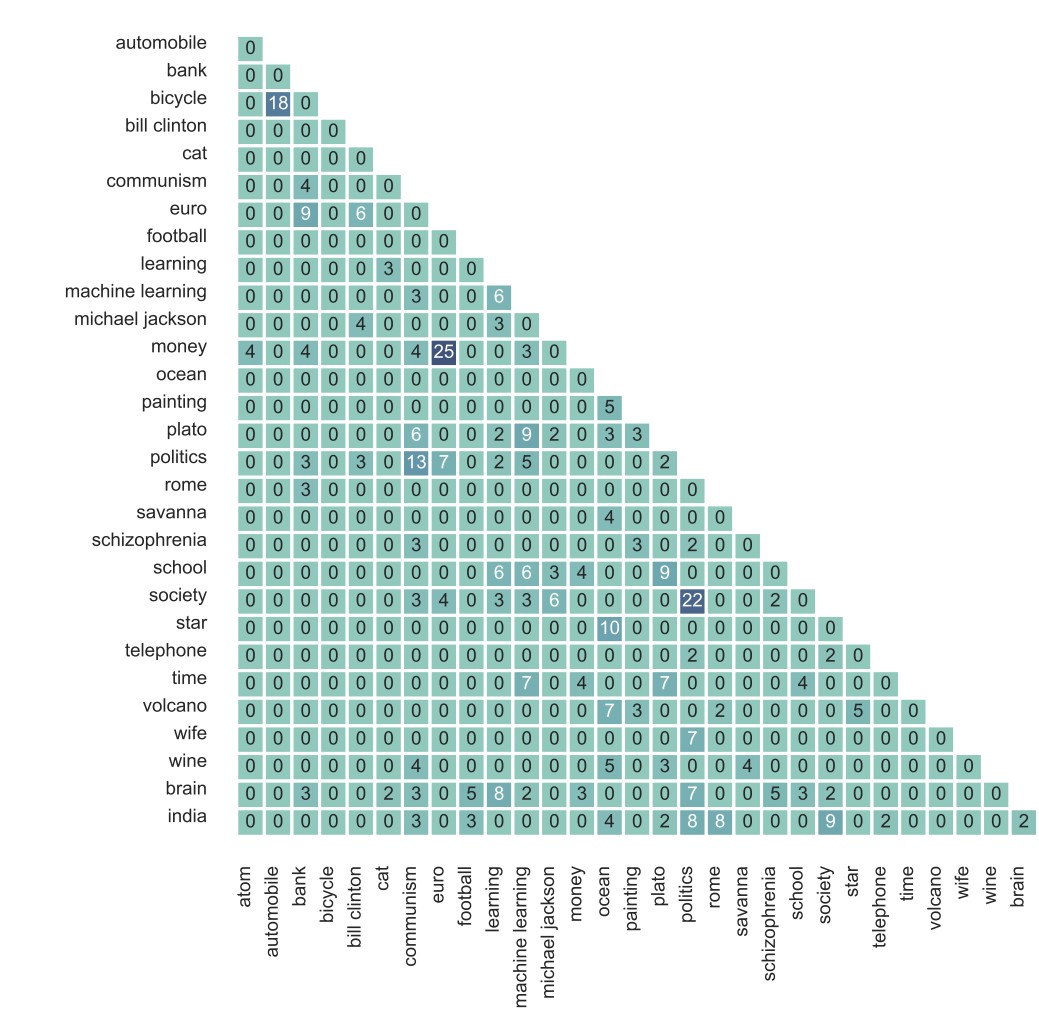

Figure 14: The overlap of all semantically relevant words across topics. $X$-axis: topic names. $Y$-axis: topic names. A number in each cell represents the percentage of semantically relevant words that are present in both topics.

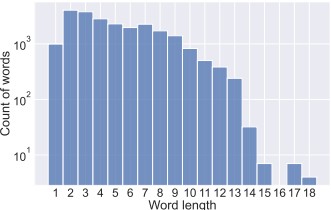

Figure 15: The distribution of word lengths in the dataset.

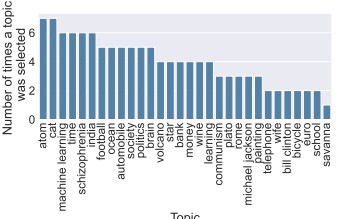

Figure 16: The number of times a topic was selected.

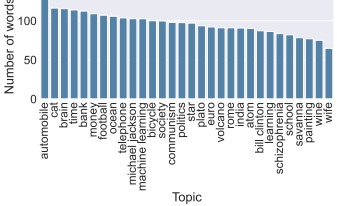

Figure 17: The number of words per topic in the dataset, as averaged across all participants.

## H.2 GENERAL GUIDELINES

1. **Focus on Word-Level Relevance:** You will receive a file containing two columns. The first column contains the words. The second column contains the names of the topics each word belongs to. No word is repeated within the same topic. The order of words within each topic

is randomised. Since the words are shuffled, evaluate each word in isolation. Context or its position within the original document does not influence the annotation. Only consider the word's relationship to the document's topic.

2. **Topic Familiarity:** Before annotating, understand the main topic of the document. Consider typical terms, concepts, and ideas associated with that topic.

3. **Function Words:** Words like articles (*a*, *the*), conjunctions (*and*, *or*), and auxiliary verbs (*is*, *are*) are generally **irrelevant**, unless they are related to the topic of a document (e.g., the topic "Function words").

4. **Domain-Specific Terms:** Technical terms or jargon specific to the topic should always be marked as **relevant**.
   *Example:* For the topic "Ocean", terms like *salinity*, *currents*, and *plankton* are relevant.

5. **Ambiguity:** If unsure about a word's relevance, use the online Cambridge English-English dictionary available at `https://dictionary.cambridge.org/dictionary/english/` to comprehend a word's meaning.

### H.3 ANNOTATION PROCEDURE

1. Please mark each word as 1 (semantically relevant) or 0 (semantically irrelevant) with respect to the topic.

2. Ensure consistent annotations throughout the task by referring to this guideline for ambiguous cases.

### H.4 EXAMPLES

**Topic: Ocean**

| Word | Annotation | Reason |
|------|------------|--------|
| ocean | 1 | Core term describing the topic. |
| saline | 1 | Describes a key property of the ocean. |
| and | 0 | Function word and unrelated to the topic. |
| beautiful | 0 | Unrelated to the topic. |
| currents | 1 | Key concept associated with the ocean. |
| divided | 0 | Unrelated to the topic. |

### H.5 QUALITY ASSURANCE

- Double-check annotations for consistency.

- Follow up with the task coordinator if further clarification is needed.

## I  DATASHEET

The datasheet can be accessed at the following URL: ANONYMOUS.

