# OpenReview forum: "An EEG dataset of word-level brain responses for semantic text relevance"
_ICLR.cc/2025/Conference — ICLR 2025 Conference Withdrawn Submission_

### Official Review · Reviewer_rwRy · 2024-10-31

**Soundness:** 2
**Presentation:** 3
**Contribution:** 2
**Rating:** 5
**Confidence:** 3

**Summary:**

This paper introduces a novel EEG dataset focused on semantic text relevance, comprising 23,270 word-level EEG recordings from participants reading texts that vary in semantic relevance to self-chosen topics. The dataset aims to bridge the gap between naturalistic reading studies and controlled psycholinguistic experiments. It provides a benchmark for evaluating models on tasks such as predicting word and sentence relevance. The dataset's contribution lies in advancing research on language relevance, psycholinguistics, and the development of brain-computer interfaces for detecting language relevance.

**Strengths:**

Originality
The manuscript presents a novel contribution by introducing an EEG dataset specifically designed for word-level semantic text correlation studies. The originality lies in its unique focus on capturing brain responses at the word level as opposed to broader natural reading or psycholinguistic paradigms commonly found in previous datasets. This is achieved through a carefully controlled experimental design where participants read sentences related or unrelated to personally chosen topics. The dataset's novelty is further emphasized by its potential to advance research in semantic processing and cognitive neuroscience.

Quality
The quality of the work is high, with meticulous attention given to the experimental setup, data collection methods, and preprocessing steps. The authors have conducted baseline experiments that include two evaluation protocols (participant-independent and participant-dependent) and two prediction tasks (lexical relevance and sentence relevance). They report benchmark performance using five established models, providing a solid foundation for future researchers to build upon.

Clarity
The manuscript is well-structured and clearly written. Definitions of terms and descriptions of methodologies are presented in a manner that is easy to follow, facilitating understanding of the dataset's construction and usage. The clarity of the document is supported by appropriate use of figures, tables, and technical language that effectively communicate the complexity of the EEG data and analysis procedures.

Significance
The significance of this work lies not only in its immediate contribution to the field of cognitive neuroscience but also in its broader implications for the development of brain-computer interfaces (BCIs) and recommendation systems that utilize neural input and feedback. By enabling the detection of language relevance online, this dataset opens up new avenues for applications in personalized learning environments, adaptive interfaces, and other contexts where real-time cognitive state monitoring could be beneficial. Moreover, the dataset has the potential to stimulate interdisciplinary research efforts, bridging gaps between neuroscience, computer science, and linguistics.

**Weaknesses:**

In the experiment, the topic switching mode that was set is fixed, and this pattern might be easily discovered by the participants. If the participants can infer that the current sentence is unrelated to the chosen topic without reading the semantic content of the sentence, this would be detrimental to the experiment. Compared to alternately presenting sentences from different themes, would a random presentation be more appropriate?

**Questions:**

In ground truth annotation, if two adjacent words only carry thematic semantics when combined, would the word level annotation for such cases be inaccurate?

---

> ### Author Response · Authors · 2024-11-21
>
> Dear reviewer rwry, thank you for your response. Here, we provide more explanations on the points that you have raised.
>
> 1. *In the experiment, the topic switching mode that was set is fixed, and this pattern might be easily discovered by the participants. If the participants can infer that the current sentence is unrelated to the chosen topic without reading the semantic content of the sentence, this would be detrimental to the experiment. Compared to alternately presenting sentences from different themes, would a random presentation be more appropriate?*
>
>     **Response:** It is a very good question that we agree requires more explanations. In the design of EEG studies, it is important to account for temporal correlations that arise from the sequential presentation of data (Li et al., 2021), which has been a serious problem in previous work (Spampinato et al., 2017). Alternating sentences from different topics effectively breaks these correlations. However, as you suggested, could a random presentation be more appropriate? While randomizing the presentation of sentences could indeed reduce temporal correlations, it introduces its own challenges. For instance, randomization might result in two or more consecutive sentences from the same topic being presented. In such cases, participants may infer from the initial words that the current sentence follows the previous sentence. If, for example, the first and second sentences from the same document are presented consecutively, the second sentence cannot be used for the evaluation of the model due to temporal correlations. Despite these potential concerns, our study's experimental design mitigates such effects. The participants were instructed to read sentences word by word while focusing on the selected topic. This setup ensured that participants semantically related each word to the topic of the document, even when the alternation order of the sentences could be guessed by the participants. The ordering of the starting document was randomised within a reading task. The existing sentence alternation strategy does not negatively impact the experiment's validity or the integrity of the results.
>
>
> 2. *In ground truth annotation, if two adjacent words only carry thematic semantics when combined, would the word level annotation for such cases be inaccurate?*
>
>     **Response**: While we acknowledge that contextual information can play a role, our annotation procedure was explicitly designed to disregard context and word position. This decision was made to minimize annotator bias, ensuring that decisions were made based solely on each word's semantic relevance in isolation, rather than influenced by the preceding word. By adopting this bag-of-words approach, we avoided potential ambiguity that could arise in cases where two adjacent words only carry semantic relevance when considered together. This strategy ensures consistency and clarity in the annotations while aligning with the study's focus on also allowing word-level analysis of semantic relevance.
>
>
>
> R. Li et al., "The Perils and Pitfalls of Block Design for EEG Classification Experiments," in IEEE Transactions on Pattern Analysis and Machine Intelligence, vol. 43, no. 1, pp. 316-333, 1 Jan. 2021
>
> C. Spampinato et al., “Deep learning human mind for automated visual classification,” in Proc. IEEE Conf. Comput. Vis. Pattern Recognit., 2017, pp. 6809–6817

---

### Official Review · Reviewer_oLJC · 2024-11-03

**Soundness:** 2
**Presentation:** 4
**Contribution:** 2
**Rating:** 5
**Confidence:** 4

**Summary:**

The paper introduces an EEG dataset collected from participants as they read sentences presented word by word on a screen. In each trial, participants chose one of two randomly selected topics from a pool of 30 and read sentences from documents related to both topics in alternating order. They were instructed to focus on their chosen topic, anticipating that they would need to explain something about it afterward, thus encouraging attention to semantically relevant words. This precise, word-by-word presentation minimizes confounding factors like eye movements, capturing time-locked EEG responses at the word level. Unlike existing datasets, this dataset uniquely captures semantic relevance in EEG signals with controlled stimuli. The authors benchmarked two tasks—word relevance prediction and sentence relevance prediction—using five well-known EEG analysis models: Linear Discriminant Analysis, Logistic Regression, EEGNet, LSTM, and UERCM. For word relevance prediction, they aimed to determine whether a word is semantically relevant to the chosen topic based on time-locked EEG epochs, with ground truth labels assigned by majority vote from three annotators. For sentence relevance prediction, they predicted whether a sentence belonged to the participant's selected topic using aggregated EEG data from its words. This dataset and initial benchmarks aim to advance research in language relevance, psycholinguistics, and the development of brain-computer interface devices for online detection of semantic relevance.

**Strengths:**

- The authors implemented a rigorous data acquisition procedure to minimize confounding factors such as eye movements and task engagement. By presenting words one at a time with consistent timing, they enhanced the reliability and validity of the EEG recordings.

- The authors benchmark experiments using five well-known models on two classification tasks (word relevance and sentence relevance) offers valuable baseline results. This enables future research to compare new models against these benchmarks.

- The paper is overall well written and easy to follow. Methodology about the data acquisition is clear and through.

**Weaknesses:**

- The dataset emphasizes word-level EEG responses, but semantic understanding in the human brain is typically constructed over larger contexts, such as sentences or entire passages. This focus on individual words may not fully capture the continuous and contextual nature of semantic processing. As a result, the dataset might have limited applicability for studying natural language comprehension in real-world settings where context plays a crucial role.

- While the authors attempted to balance topic representation, the dataset still exhibits inconsistencies because participants selected different topics to focus on during the trials. Since each participant was presented with a unique combination of topics and chose their preferred topic from randomly selected pairs, the exposure to specific topics varies across participants. This variability can introduce inconsistencies in the dataset, as some topics may have more data associated with them than others, and participants may not have been equally distributed across all topics. To address this issue, the authors could have standardized the topic selection process to ensure that all participants are exposed to the same set of topics, or at least to a balanced and representative subset.

- The annotation process for determining word-level semantic relevance does not seem appropriate. The guidelines provided to annotators were not detailed enough, which may have led to inconsistent labeling—especially for words like "contains" and "stated" (line 250) that can be relevant depending on context. Moreover, with only three annotators involved, there's a higher risk that personal biases could influence the results. To improve the reliability and consistency of the annotations, the authors should offer more detailed, context-aware guidelines to the annotators and consider increasing the number of annotators to reduce individual biases. Incorporating systematic methods, such as semantic similarity measures, could also help validate and refine the annotations, making the dataset more robust.

- The classification results (table 3), particularly in the participant-independent setting, are only slightly better than chance levels. Precision scores hover around 51–57%, which, given the class imbalance, indicates limited predictive power. This modest performance questions the practical utility of the models and suggests that decoding semantic relevance from EEG signals remains a significant challenge with the current approach/dataset. The authors could have explored more state-of-the-art models for their classification task or did some ablation studies and tune the hyperparameters for their training for the optimal result on their dataset.

- One area for improvement could be adding statistical significance testing to determine whether the observed results are meaningful or could have occurred by chance. This would make it easier to assess/compare the true effectiveness of the proposed models.

**Questions:**

-The LDA and LR models perform worse in the participant-dependent setting compared to the participant-independent setting, which is counterintuitive. Typically, models fine-tuned on individual participants' data should perform better due to personalization.  What could explain this?

- Although the authors state that the dataset and code are openly released, they were not made accessible during the review process. Is it possible to share it as a supplementary material for the sake of the review process?

**Details Of Ethics Concerns:**

While the paper mentions ethical considerations, it is important to proceed with caution when making participants' neural data publicly available.

---

> ### Author Response · Authors · 2024-11-21
>
> Dear reviewer oljc, thank you for your response. Here, we provide our response to the points that you have raised.
>
> 1. *The dataset emphasizes word-level EEG responses, but semantic understanding in the human brain is typically constructed over larger contexts, such as sentences or entire passages. This focus on individual words may not fully capture the continuous and contextual nature of semantic processing. As a result, the dataset might have limited applicability for studying natural language comprehension in real-world settings where context plays a crucial role.*
>
>     **Response**: This is a misunderstanding. Our data is not limited to word-level responses. Although the words were presented individually for approximately 0.7 seconds on the screen, the EEG recordings are continuous, capturing neural responses not only at the word level, but also across entire sentences and even between sentences as participants read sequentially. While we provide word-level responses to mitigate confounding factors such as eye movement artefacts (Hollenstein et al., 2018), our dataset enables the study of semantic relevance at multiple levels, including word, sentence, and cross-sentence contexts. Therefore, we believe our dataset is well suited for studying natural language comprehension, including the contextual aspects of semantic processing.
>
> 2. *While the authors attempted to balance topic representation, the dataset still exhibits inconsistencies because participants selected different topics to focus on during the trials. Since each participant was presented with a unique combination of topics and chose their preferred topic from randomly selected pairs, the exposure to specific topics varies across participants. This variability can introduce inconsistencies in the dataset, as some topics may have more data associated with them than others, and participants may not have been equally distributed across all topics. To address this issue, the authors could have standardized the topic selection process to ensure that all participants are exposed to the same set of topics, or at least to a balanced and representative subset.*
>
>     **Response**: The complete balancing of the topics among the participants was not feasible due to the experimental design, which required the participants to choose one of the two topics they wanted to learn more about. This approach ensures that the participant's intrinsic interest or preference is prioritised rather than being influenced by enforced balancing. Figure 16 in the Appendix illustrates the frequency of topic selections among participants. Importantly, the unbalanced distribution of selected topics does not introduce inconsistencies in the dataset. For each reading task, one topic was selected and the other was not, establishing semantic relevance in an absolute manner. By “absolute”, we mean that relevance is determined solely by the relationship between the selected topic and the unselected one, independent of other topics. Additionally, Figure 17 in the Appendix provides a breakdown of the number of words per topic. While some topics may have more words than others, we do not anticipate that this unbalanced distribution introduces inconsistencies in the dataset. We have added the above information to the Section C.3 in the Appendix (LINES: 872 - 884).
>
> 3. *The annotation process for determining word-level semantic relevance does not seem appropriate. The guidelines provided to annotators were not detailed enough, which may have led to inconsistent labeling—especially for words like "contains" and "stated" (line 250) that can be relevant depending on context. Moreover, with only three annotators involved, there's a higher risk that personal biases could influence the results. To improve the reliability and consistency of the annotations, the authors should offer more detailed, context-aware guidelines to the annotators and consider increasing the number of annotators to reduce individual biases. Incorporating systematic methods, such as semantic similarity measures, could also help validate and refine the annotations, making the dataset more robust.*
>
>     **Response**: We include the complete Guidelines in the Section H of the Appendix (LINES: 1391 - 1403, 1454 - 1496). With regard to you comment about the number of annotators, it was shown that three annotators are sufficient to have consistent performance and adding more annotators results only in minor improvements in performance  (Wilm et al., 2021). Therefore, we believe that adding more annotators will not change significantly the resulted annotations. We have added this information into the first paragraph of the Section 4.2 (LINES: 241 - 244). As per your suggestion, we will add a semantic similarity measure additionally to the provided annotations and will include the results in the Appendix for the camera-ready version of the paper. Thank you for pointing this out.

---

> ### Author Response · Authors · 2024-11-21
>
> 4. *The classification results (table 3), particularly in the participant-independent setting, are only slightly better than chance levels. Precision scores hover around 51–57\%, which, given the class imbalance, indicates limited predictive power. This modest performance questions the practical utility of the models and suggests that decoding semantic relevance from EEG signals remains a significant challenge with the current approach/dataset. The authors could have explored more state-of-the-art models for their classification task or did some ablation studies and tune the hyperparameters for their training for the optimal result on their dataset.*
>
>
>     **Response**: Participant-independent evaluation is significantly more challenging than participant-dependent evaluation, which may explain the difference in performance between the two strategies. Participant-independent classification has been shown to be particularly difficult in the domain of brain recordings (Hang et al., 2021; Lawhern et al., 2018), as brain responses to the same stimuli can vary substantially among participants.
>
>     The issue of class imbalance applies only to the word relevance classification task, while the data for the sentence relevance classification task is balanced. In our benchmark experiments, we used machine learning models that already outperform previously reported results on similar tasks (Hollenstein et al., 2018; Ye et al., 2022). This improved performance demonstrates that our dataset is valuable for advancing research, and it provides a robust platform for evaluating novel machine learning models. The five models included in our study establish baseline performance metrics, offering researchers a reference point for future developments and comparisons.
>
>
> 5. *One area for improvement could be adding statistical significance testing to determine whether the observed results are meaningful or could have occurred by chance. This would make it easier to assess/compare the true effectiveness of the proposed models.*
>
>
>     **Response**: We did not include the statistical significance testing as the results are statistically significant throughout when tested against the control models trained with permuted labels. Statistical significance was assessed for the reported AUC scores using the non-parametric bootstrapping method with 10000 permutations, considering a $p$-value threshold of <0.01. In the camera-ready version of the paper (the first paragraph of the Section 5; the Results paragraph of the Section 5.4; the Results paragraph of the Section 5.5), we will describe how the statistical significance testing was performed and include the results of statistical significance testing.
>
>
> 6. *The LDA and LR models perform worse in the participant-dependent setting compared to the participant-independent setting, which is counterintuitive. Typically, models fine-tuned on individual participants' data should perform better due to personalization. What could explain this?*
>
>     **Response**: It is a very good question that we agree was not clear in the paper. In the participant-dependent setting, the LDA and LR models are trained from scratch, as these models do not support fine-tuning in the conventional sense. Consequently, they lack sufficient data in this setting to learn robust and generalizable features, resulting in lower performance compared to the participant-independent setting, where a larger and more diverse dataset is available for training.
>
>     This limitation is consistent with findings in the literature. For example, Hang et al. (2021) also reported cases in which models trained on limited participant-specific data failed to outperform generic classifiers. Thus, the observed results are likely due to the restricted data availability in the participant-dependent strategy.
>
>     In Section E.6 of the Appendix (LINES: 1330 - 1339) we have added the above explanations. Additionally, in the Results paragraph of the Section 5.5 of the paper, we have added text that points to E.6 for a detailed explanation. In the Table 3, we have added a note that the LDA and LR models are trained from scratch in the within-subject paradigm, as fine-tuning is not supported for these models.
>
> 7. *Dataset and code availability*
>
>     **Response**: The dataset (anonymized) and code (anonymized) can be accessed here: https://huggingface.co/datasets/Quoron/EEG-semantic-text-relevance.

---

> ### Author Response · Authors · 2024-11-21
>
> As the terms “within-subject” and “cross-subject” are more commonly used, we have replaced “participant-dependent” and “participant-independent” with “within-subject” and “cross-subject”, respectively.

---

### Official Review · Reviewer_vE9o · 2024-11-03

**Soundness:** 3
**Presentation:** 3
**Contribution:** 3
**Rating:** 6
**Confidence:** 4

**Summary:**

The paper releases an EEG dataset where the participants perform a Rapid Serial Visualisation Task, reading the words relevant/irrelevant to a preselected topic from Wikipedia. The work later benchmarks the performance of relevant word/sentence prediction from the EEG activity recorded during the task from 15 participants using 5 different prediction models in participant-dependent and independent scenarios.

**Strengths:**

The work plans to release the first public dataset for ERP analysis of semantic relevance in a RSV paradigm. The experimental paradigm and data collection is described with fair clarity. The dataset can significantly contribute to the domain as it opens up opportunities to build novel methods and benchmark performance on such tasks.

The work includes ERP analysis to validate the data with previous insights from literature and benchmarks 5 different architectures/models in both participation-independent and dependent scenarios.

**Weaknesses:**

It would be helpful to clearly define the novelty involved with this work. It is understood that while similar paradigms have been used before with different topics for the ERP analysis as cited by the authors, techniques/models are well known, novelty is limited to dataset availability and benchmark.

The authors cite the limitation of the benchmark in terms of the models used that do not capture spatio-temporal features from the data. However, there are models available that have addressed such needs in recent literature, and a better state-of-the-art benchmark is a possibility. Authors may refer to :
- STNet: https://torcheeg.readthedocs.io/en/v1.1.0/generated/torcheeg.models.STNet.html
Other models using Riemannian geometry:
- https://github.com/CECNL/MAtt
- https://github.com/zhiwu-huang/SPDNet

**Questions:**

While the work is explained with fair clarity, a few aspects need to be addressed:
1. The experimental paradigm can be explained with a diagram indicating the inter-stimulus interval. Authors mention each epoch as -200ms to 1000ms and later use 250ms to 950ms data for classification. It is confusing for the reader. This gives the impression that 200ms from previous trials/stimuli has been considered as the baseline.

2. It would be helpful to explain the shortest and the longest word and the font size. It is known that visual cues, including font size and style, impact ERP components as far as N400. While the work ignores the first 250ms, checking visual cues is also important.

3. For Figure 2a and 2b, the findings to be validated against insights from previous literature can be better represented with selective channels. As per the reference Eugster et al 2016:  "initial fronto-central positivity at 300 ms, relative to the onset of the word on the screen, followed by a centro-parietal positivity from 400 to 600 ms." The topography in Fig 2a and 2b are challenging to comprehend, while the figures in the appendix are a bit easier to comprehend and compare.

4. The applications' significance and outcomes can be elaborated in the discussion for the readers.

**Details Of Ethics Concerns:**

While the concern is not alarming, since the authors mention the ethics codes followed to collect the data, the dataset links are not shared to maintain anonymity. Therefore, a check is necessary if all the due codes are followed to protect the privacy of the human subjects during data release.

---

> ### Author Response · Authors · 2024-11-21
>
> Dear reviewer vE9o, thank you for your response. Here, we provide our response to the points that you have raised.
>
> 1.	*It would be helpful to clearly define the novelty involved with this work. It is understood that while similar paradigms have been used before with different topics for the ERP analysis as cited by the authors, techniques/models are well known, novelty is limited to dataset availability and benchmark.*
>
>     **Response**: Thank you for your comment. The novelty of our work extends beyond dataset availability and benchmark results. The novelty of our work lies in specifically capturing semantic text relevance through time-locked word presentation. This distinguishes our dataset from those proposed by Hollenstein et al. (2018, 2020), who did not use time-locked EEG acquisition, and by Ye et al. (2022), whose dataset focused on brain responses during a question-answering task. We have added the above information about the novelty to the Introduction and Conclusion sections to strengthen the clarity of contributions (LINES: 047 - 048, 520-522).
>
>
> 2. *The authors cite the limitation of the benchmark in terms of the models used that do not capture spatio-temporal features from the data. However, there are models available that have addressed such needs in recent literature, and a better state-of-the-art benchmark is a possibility.*
>
>     **Response**: To clarify, the EEGNet, LSTM, and UERCM models do indeed account for spatio-temporal features in the EEG recordings. Specifically, EEGNet is explicitly designed to capture these features via convolutional network architecture. For the LSTM model, temporal dependencies are captured via hidden and cell states and the spatial features are embedded into the vectors of data that represent the recorded EEG responses for a specific time step. Similarly, the UERCM model captures temporal and spatial relationships via self-attention. We have revised the paragraph in the Discussion section (LINES: 486 - 497) to enhance clarity and provide a sharper explanation, referencing the research you mentioned to facilitate a better understanding.
>
> 3. *The experimental paradigm can be explained with a diagram indicating the inter-stimulus interval. Authors mention each epoch as -200ms to 1000ms and later use 250ms to 950ms data for classification. It is confusing for the reader. This gives the impression that 200ms from previous trials/stimuli has been considered as the baseline.*
>
>     **Response**: We will add the diagram to the camera-ready version of the paper in the Appendix. This diagram will clarify the experimental design, showing (1) word presentation time corresponding to 700ms, (2) epochs spanning from -200ms to 1000ms, (3) responses from 250ms to 950ms used for machine learning experiments. As the [-200 ms, 0 ms] range is relative to stimulus onset, it means that -200ms is taken from the previous 1000 ms sample. Thus, the [800 ms, 1000 ms] range is the baseline from previous presentation. This strategy allows analysis that are independent of other samples (i.e. avoid computing a mean baseline from the entire dataset, which would be unrealistic for many realistic applications). We will add the above explanation to the Section C.2 of the Appendix.

---

> ### Author Response · Authors · 2024-11-21
>
> 4. *It would be helpful to explain the shortest and the longest word and the font size. It is known that visual cues, including font size and style, impact ERP components as far as N400. While the work ignores the first 250ms, checking visual cues is also important.*
>
>     **Response**: We have added a new Figure 15 in the Appendix that shows the distribution of word lengths. The length of the longest word is 18 and the length of the shortest word is 1. Font size is 18 pt, as described already in Section 3.2. In our benchmark machine learning experiments, we consider EEG recordings within the 250-950 ms. The selection of this time range is based on neurolinguistic research showing that ERPs occurring 250 to 700 ms after stimulus perception are likely indicators of the relevance of language stimuli (Kim and Osterhout, 2005), meaning that recordings prior to 250 ms are insignificant for the present stimuli. We specifically did not want the visual potentials to affect the results. For example, we would expect different responses between 0-250 ms based on word length as more photons (light) on the screen would evoke elevated potentials during this temporal segment. However, this is not a factor that we want to measure as word length is not a factor we want to account for. Instead, we wanted to make sure our ERP effects only account for relevance and semantic processing of stimuli (independent of how much light on the screen their presentation requires). Therefore, on purpose, the 0-250 ms data were ignored. We have added the above information to the Section 5.3 (LINES: 372 - 380).
>
> 5. *For Figure 2a and 2b, the findings to be validated against insights from previous literature can be better represented with selective channels. As per the reference Eugster et al 2016: "initial fronto-central positivity at 300 ms, relative to the onset of the word on the screen, followed by a centro-parietal positivity from 400 to 600 ms." The topography in Fig 2a and 2b are challenging to comprehend, while the figures in the appendix are a bit easier to comprehend and compare.*
>
>     **Response**: Thank you for the suggestion. The statistical ERP analysis presented in Section 4.3 confirms that the ERP results are consistent with Eugster et al. (2016). We acknowledge that Figure 2 may require careful observation to interpret. To improve clarity, we have added a new Figure 9 in the Appendix, which illustrates the differences between ERP responses for semantically relevant and those for semantically irrelevant words and, as described in Section C.4 in the Appendix (LINES: 889 - 895), shows that the patterns of these differences align with the positivity patterns for the components P300, N400, and P600 reported by Eugster et al. (2016). Additionally, a new Figure 10 in the Appendix highlights that the differences in ERP responses for the Fz, C3, C4, P3, Pz, and P4 electrodes between semantically relevant and semantically irrelevant words are statistically significant $(p < 0.001)$. We have added a text to the caption of the Figure 2 in the paper that refers to Figures 5, 6, 7, and 8 which compare ERPs between semantically relevant and semantically irrelevant words for each electrode separately.
>
> 6. *The applications' significance and outcomes can be elaborated in the discussion for the readers.*
>
>     **Response**: We have elaborated more on this in the Discussion section (LINES: 500 - 506). Specifically, our dataset can enable applications in assistive technologies, such as adaptive learning systems and personalised content delivery, based on user engagement and interest. Moreover, our benchmark experiments highlight the potential of machine learning models to decode semantic relevance from EEG data, offering a foundation for extending these capabilities to new application domains like brain-state driven entertainment, neurofeedback training (training memory and attention through brain-relevance feedback), and cognitive workload monitoring to optimise task assignment and performance.
>
> Kim, A., Osterhout, L. The independence of combinatory semantic processing: Evidence from event-related potentials. Journal of Memory and Language, 52(2), 205-225, 2005.

---

### Official Review · Reviewer_zbc2 · 2024-11-03

**Soundness:** 2
**Presentation:** 2
**Contribution:** 2
**Rating:** 3
**Confidence:** 5

**Summary:**

The study introduces a new EEG dataset collected from 15 participants engaged in reading tasks with language processing requirement, designed to support NLP research in areas such as word and sentence relevance classification, while EEG signal can be cogntive GT. This dataset further supports exploring neural representations by complementing existing EEG-based reading tasks.

**Strengths:**

This study creates a new EEG-based dataset specifically designed for NLP tasks. Such a dataset is instrumental in advancing research on BCIs for decoding language processing for reading which could enhance our understanding and modeling of neural representations and expand to text extended-multimodality applications in the future.

**Weaknesses:**

The participant background is unclear; the authors briefly mention “high English fluency,” this indicates participants are non-native English speakers, so that standardized tests like IELTS or TOEFL would provide a more objective assessment of their reading proficiency, as language processing and brain activity can vary significantly between native and non-native speakers.

Regarding the dataset’s scale, while the total number of events and the recording duration are appreciated, the sample size of 15 participants appears comparable to, or even smaller than, existing studies, which may limit generalizability.

The use of time-locked event recording could potentially induce stress in participants, particularly for non-native speakers (see my first concern). An analysis of the effects of this recording approach on participant responses would add depth to the study, possibly by examining trial-level data.

In EEG-based BCI studies, terms like “within-subject” and “across-subject” are more commonly used than “participant-dependent” and “participant-independent.”

The intended purpose of this dataset could be more explicitly defined for ML-based classification (for section 5). Authors shall show that their dataset addresses the accuracy limitations observed in prior datasets. Studies like ZuCo have already demonstrated promising results in word relevance classification tasks, and a comparison or advancement upon those findings would strengthen the contribution of this dataset.

**Questions:**

See the above and the additional one below.

The ERP analysis lacks some clarity. Typical stimuli in previous EEG-based NLP research have been shown to elicit P300/N400 responses. It would be helpful to understand whether this study captures ERP responses across all words or only specific examples. Ideally, the study would present averaged or comprehensive ERP potentials for all words, with semantic labels serving as ground truth.

---

> ### Author Response · Authors · 2024-11-21
>
> Dear reviewer zbc2, thank you for your comments. Here is our response.
>
> 1. *The participant background is unclear; the authors briefly mention “high English fluency', this indicates participants are non-native English speakers, so that standardized tests like IELTS or TOEFL would provide a more objective assessment of their reading proficiency, as language processing and brain activity can vary significantly between native and non-native speakers.*
>
>     **Response**: All participants were volunteers from universities, meaning they were either students or staff members. Regarding their English proficiency, participants demonstrated high fluency, with an average score of 23.53 (SD = 1.23) on a standardised English proficiency test (Cambridge English Language Assessment, 2024), where the maximum possible score is 25. This score reflects strong English language skills, supporting the claim of “high English fluency”. We believe that the used test provides a reliable indication of participants' reading proficiency. We have added the above information to the Section 3.1 (LINES: 106-107, 133-134). More generally, Cambridge English test is recognized by 25000 organizations in 130 countries around the world. It is accepted by more than 100 universities in the US, and 99\% of the 100 best universities around the world featured in the Times Higher University Ranking.
>
> 2. *Regarding the dataset’s scale, while the total number of events and the recording duration are appreciated, the sample size of 15 participants appears comparable to, or even smaller than, existing studies, which may limit generalizability.*
>
>     **Response**: A sample size of 15 participants is consistent with the sample sizes used in comparable EEG studies, such as those by Ye et al. (2022) (21 participants, 465 sentences, around 4600 words) and Hollenstein et al. (2018) (12 participants, 407 sentences, 8164 words). Our dataset contains 23,270 time-locked EEG recordings, providing a substantial amount of data for model training and testing. Brysbaert (2019) demonstrates that, with well-designed experiments, even small participant pools can provide sufficient generalizability. Given that relevance is often subjective and may manifest in brain responses differently for different participants, we also think it is more important to have more samples per participant than data from more participants. Thus, we believe that our rigorously designed EEG data collection procedure, which carefully controlled for ordering effects and confounding factors, ensures that the dataset is robust and generalizable for its intended tasks. We have added the above information to the Discussion section (LINES: 473-485).
>
>
> 3. *The use of time-locked event recording could potentially induce stress in participants, particularly for non-native speakers (see my first concern). An analysis of the effects of this recording approach on participant responses would add depth to the study, possibly by examining trial-level data.*
>
>     **Response**: In our dataset, where each word is presented for approximately 0.7 seconds, we use principles similar to those underlying Rapid Serial Visual Presentation (RSVP) to ensure precise time-locked EEG recordings. RSVP is an effective method for collecting data in studies that require precise temporal alignment between stimuli and acquired data (Potter, 1984). A search for "rapid serial visual presentation" and "EEG" in Google Scholar for 2023 yielded 508 results, demonstrating its broad application in the field. We have reflected on this in the Section B of the Appendix (LINES: 826 - 833).
>
>     Regarding the potential stress induced by RSVP, particularly for non-native speakers, research suggests that the RSVP does not significantly affect text comprehension between native and non-native speakers when reading at speeds of 1000 words per minute (wpm) and 500 wpm during RSVP tasks (Zann and Conklin, 2015). Notably, we use a slower speed presenting each word for around 0.7s (around 86 wpm).
>
> Cambridge English Language Assessment. Test your English – General English. http://www.cambridgeenglish.org/test-your-english/adult-learners/. Accessed: 17.11.2024.
>
> Boo, Zann and Kathy Conklin. “The impact of Rapid Serial Visual Presentation (RSVP) on reading by nonnative speakers.” Journal of Language Teaching and Research 4 (2015): 111-129.

---

> ### Author Response · Authors · 2024-11-21
>
> 4. *In EEG-based BCI studies, terms like “within-subject” and “across-subject” are more commonly used than “participant-dependent” and “participant-independent.”*
>
>     **Response**: Thank you for pointing this out. We assume you mean "cross-subject" instead of  "across-subject" (Lawhern et al., 2016; Kingphai et al., (2024)). Although, participant-dependent/independent and within/cross-subject are all commonly used in the literature -- as per your suggestion, we have replaced the terms “participant-dependent” and “participant-independent” with “within-subject” and “cross-subject”, respectively.
>
> 5. *The intended purpose of this dataset could be more explicitly defined for ML-based classification (for section 5). Authors shall show that their dataset addresses the accuracy limitations observed in prior datasets. Studies like ZuCo have already demonstrated promising results in word relevance classification tasks, and a comparison or advancement upon those findings would strengthen the contribution of this dataset.*
>
>     **Response**: Thank you for your insightful suggestion on strengthening the contribution of our dataset. The purpose of our dataset is to capture semantic text relevance, a task not previously addressed, but also to demonstrate the advantages of time-locked EEG acquisition. Specifically, while the ZuCo dataset uses eye tracking to time-lock EEG recordings, our approach employs time-locked word presentation, which helps avoid the confounding effects of eye movement artefacts and more precise timing. Zhang et al. (2024) presented results in word relevance classification tasks, in which they generated the ground-truth labels for words using LLMs. They report accuracy scores of around 60\%, which is in line with previous research (Eugster et. al., 2014; Jacucci et al., 2019). Therefore, we see limited benefits of using eye-locked potentials as they only risk involving confounded data (Thielen et al., 2019). Furthermore, eye-movements are well-known to vary along semantic dimensions, for example, complex sentences result in frequent recurrent movements and unexpected, longer, and unfamiliar words increase fixation durations (Metzner et al., 2017).  We will add the above elaboration on the comparison with the ZuCO dataset to the Discussion section of the camera-ready version of the paper to better highlight these comparisons and further clarify the contributions of our dataset.
>
> 6. *The ERP analysis lacks some clarity. Typical stimuli in previous EEG-based NLP research have been shown to elicit P300/N400 responses. It would be helpful to understand whether this study captures ERP responses across all words or only specific examples. Ideally, the study would present averaged or comprehensive ERP potentials for all words, with semantic labels serving as ground truth.*
>
>     **Response**: We apologize for the confusion, which we have now addressed in the paper. The conducted ERP analysis is performed across all words and participants for the words that are annotated as semantically relevant and semantically irrelevant, therefore matching directly to the ground truth labels. The statistical analysis shows that there are significant differences in relevance effect between semantically relevant and semantically irrelevant words for P300, N400, and P600. Additionally, Figures 5, 6, 7, and 8 show ERPs for each electrode averaged across all words that were annotated as semantically relevant and semantically irrelevant. We will improve the description of the ERP analysis in Section 4.3.
>
> K. Kingphai and Y. Moshfeghi, "Mental Workload Assessment Using Deep Learning Models from EEG Signals: A Systematic Review," in IEEE Transactions on Cognitive and Developmental Systems, 2024.
>
> Zhang Y, et al. Integrating Large Language Model, EEG, and Eye-Tracking for Word-Level Neural State Classification in Reading Comprehension. IEEE Trans Neural Syst Rehabil Eng. 2024. doi: 10.1109/TNSRE.2024.3435460.
>
> Eugster, M. J. A., Ruotsalo, T., Spapé, M. M., Kosunen, I., Barral, O., Ravaja, N., Jacucci, G., & Kaski, S. (2014). Predicting term-relevance from brain signals. SIGIR 2014 - Proceedings of the 37th International ACM SIGIR Conference on Research and Development in Information Retrieval, 425–434. https://doi.org/10.1145/2600428.2609594
>
> Jacucci, G., et al. (2019). Integrating neurophysiologic relevance feedback in intent modeling for information retrieval. Journal of the Association for Information Science and Technology, 70(9), 917–930.
>
> Thielen, J., Bosch, S.E., van Leeuwen, T.M. et al. Evidence for confounding eye movements under attempted fixation and active viewing in cognitive neuroscience. Sci Rep 9, 17456 (2019).
>
> Metzner, P., et al. (2017), The Importance of Reading Naturally: Evidence From Combined Recordings of Eye Movements and Electric Brain Potentials. Cogn Sci, 41: 1232-1263. https://doi.org/10.1111/cogs.12384

---

> ### Author Response · Authors · 2024-11-28
>
> Thank you for your comment.
>
> Here are our responses.
>
>
> 1. *"most of the work uses the ZuCo dataset for word-level recognition" from your last comment and "Studies like ZuCo have already demonstrated promising results in word relevance classification tasks" from your previous comment"*
>
> **Response**: In our previous response numbered 5, we referred to the sentence classification tasks using word-level brain responses. This is a misunderstanding, and we have now edited the previous comment focusing on word relevance classification tasks. The previous comment now contains the following information. Zhang et al. (2024) presented results in word relevance classification tasks, in which they generated the ground-truth labels for words using LLMs. They report accuracy scores of around 60\%, which is in line with previous research (Eugster et. al., 2014; Jacucci et al., 2019). Therefore, we see limited benefits of using eye-locked potentials as they only risk involving confounded data (Thielen et al., 2019). Furthermore, eye-movements are well-known to vary along semantic dimensions, for example, complex sentences result in frequent recurrent movements and unexpected, longer, and unfamiliar words increase fixation durations (Metzner et al., 2017). Overall, this makes the eye-tracking method for time-locking less reliable for recording accurate data.
>
> We will add the above elaboration on the comparison with the ZuCO dataset to the Discussion section of the camera-ready version of the paper to better highlight these comparisons and further clarify the contributions of our dataset as well as limitations and potential confounds of alternative datasets.
>
> 2. *"time-lock design, it is not a natural reading task; it is simply aligning or mapping words to EEG signals without semantic intentions,"*
>
> **Response**: Our aim was to imitate natural reading, as written in the paper. Word-by-word presentation is known to closely resemble "natural reading" (Li et al., 2022; Mäkelä 2024, Wang et al., 2024). Although we agree that single word reading is not nearly as natural as "free" reading, we ensured that participants were able to process the information at the sentence level by asking them to describe the contents of the article at the end of each block and these summaries were congruent with the presented information.
>
> We disagree that the time-locked design of our experiment in which sentences were presented one word at a time and word after word does not consider semantic intentions. Research using word-by-word presentation has shown that semantic intentions are captured when using a word-by-word presentation (Ye et al., 2007; Kutas et al., 1984; Kutas et al., 2011). EEG/ERP analysis with rapid serial visual presentation has also been the cornerstone of human semantic language processing, starting from the classic studies where P300, N400 and P600 effects were found. These are the main semantic effects associated with human natural language processing that can be recorded via EEG and they have been studied by using word-by-word presentation (Martín-Loeches et al., 2017; Barber et al., 2010).
>
> Additionally, in "natural-reading" paradigms that rely on eye movements, as in the ZuCo dataset, artefacts (recurring gaze movement, fixation duration, etc.) systematically covary with semantic integration and other processes in language comprehension. To avoid this serious confound, we therefore opted for the sequential word presentation paradigm.
>
>
>
> 3. "non-native English speakers may feel less confident"
>
> **Response**: The Wikipedia documents belong to general topics such as Ocean, Cat, and Euro and we used only the first six sentences. In our previous comment, we showed that participants have high English proficiency supported by the research that RSVP does not significantly affect text comprehension between native and non-native speakers (Zann and Conklin, 2015). Furthermore, point 2. of this comment references literature indicating that a word-by-word presentation emulates "natural reading" and comprehensively accounts for semantic information. Therefore, we do not think that non-native English speakers may feel less confident due to a word-by-word presentation.

---

> ### Author Response · Authors · 2024-11-28
>
> References related to the previous comment:
>
> Zhang Y, Li Q, Nahata S, Jamal T, Cheng SK, Cauwenberghs G, Jung TP. Integrating Large Language Model, EEG, and Eye-Tracking for Word-Level Neural State Classification in Reading Comprehension. IEEE Trans Neural Syst Rehabil Eng. 2024;32:3465-3475. doi: 10.1109/TNSRE.2024.3435460. Epub 2024 Sep 20. PMID: 39141467.
>
> Thielen, J., Bosch, S.E., van Leeuwen, T.M. et al. Evidence for confounding eye movements under attempted fixation and active viewing in cognitive neuroscience. Sci Rep 9, 17456 (2019). https://doi.org/10.1038/s41598-019-54018-z
>
> Ye Z, Zhan W, Zhou X. The semantic processing of syntactic structure in sentence comprehension: an ERP study. Brain Res. 2007 Apr 20;1142:135-45. doi: 10.1016/j.brainres.2007.01.030. Epub 2007 Jan 18. PMID: 17303093.
>
> Kutas M, Hillyard SA. Brain potentials during reading reflect word expectancy and semantic association. Nature. 1984 Jan 12-18;307(5947):161-3. doi: 10.1038/307161a0. PMID: 6690995.
>
> Kutas M, Federmeier KD. Thirty years and counting: finding meaning in the N400 component of the event-related brain potential (ERP). Annu Rev Psychol. 2011;62:621-47. doi: 10.1146/annurev.psych.093008.131123. PMID: 20809790; PMCID: PMC4052444.
>
> Martín-Loeches, M., Ouyang, G., Rausch, P., Stürmer, B., Palazova, M., Schacht, A., Sommer, W. (2017). Test–retest reliability of the N400 component in a sentence-reading paradigm. Language, Cognition and Neuroscience, 32(10), 1261–1272. https://doi.org/10.1080/23273798.2017.1330485
>
> Barber HA, Doñamayor N, Kutas M, Münte T. Parafoveal N400 effect during sentence reading. Neurosci Lett. 2010 Jul 26;479(2):152-6. doi: 10.1016/j.neulet.2010.05.053. Epub 2010 May 24. PMID: 20580772; PMCID: PMC4096702.
>
> Li C, Midgley KJ, Holcomb PJ. ERPs Reveal How Semantic and Syntactic Processing Unfold across Parafoveal and Foveal Vision during Sentence Comprehension. Lang Cogn Neurosci. 2023;38(1):88-104. doi: 10.1080/23273798.2022.2091150. Epub 2022 Jun 23. PMID: 36776698; PMCID: PMC9916175.
>
> Metzner, P., von der Malsburg, T., Vasishth, S. and Rösler, F. (2017), The Importance of Reading Naturally: Evidence From Combined Recordings of Eye Movements and Electric Brain Potentials. Cogn Sci, 41: 1232-1263. https://doi.org/10.1111/cogs.12384
>
> Mäkelä, Sasu, Challenges in magnetoencephalographic studies of naturalistic reading and speech, 2024. https://aaltodoc.aalto.fi/items/d1394262-843f-4365-8bf7-1c6e8fb013a5
>
> Wang, L., Frisson, S., Pan, Y., Jensen, O. (2024). Fast hierarchical processing of orthographic and semantic parafoveal information during natural reading. https://doi.org/10.1101/2024.09.27.615440
>
> Eugster, M. J. A., Ruotsalo, T., Spapé, M. M., Kosunen, I., Barral, O., Ravaja, N., Jacucci, G., Kaski, S. (2014). Predicting term-relevance from brain signals. SIGIR 2014 - Proceedings of the 37th International ACM SIGIR Conference on Research and Development in Information Retrieval, 425–434. https://doi.org/10.1145/2600428.2609594
>
> Jacucci, G., Barral, O., Daee, P., Wenzel, M., Serim, B., Ruotsalo, T., Pluchino, P., Freeman, J., Gamberini, L., Kaski, S., Blankertz, B. (2019). Integrating neurophysiologic relevance feedback in intent modeling for information retrieval. Journal of the Association for Information Science and Technology, 70(9), 917–930. https://doi.org/10.1002/asi.24161

---

### Author Response · Authors · 2024-11-16
**Dataset availability**

Dear Reviewers,

Thank you for your comments.

We will provide our responses as soon as possible.

Some of you asked whether the dataset can be accessed.
We apologize for not providing the links to the anonymized version of the dataset and code earlier.

The dataset (anonymized) and code (anonymized) can be accessed here: https://huggingface.co/datasets/Quoron/EEG-semantic-text-relevance.

Kind regards,

The Authors

---

### Author Response · Authors · 2024-11-21

Dear reviewers,

thank you for your detailed reviews and valuable suggestions to improve the paper. After reading the reviews we did not identify any critical issues, but rather clarified some misunderstandings and added many missing details here and in the paper. As most of the scores were borderline, we hope you could go through our responses and see if you would be willing to raise your scores in response to our revisions and clarifications.

We have responded to each of you separately.

---

### Note · Authors · 2025-01-23

I have read and agree with the venue's withdrawal policy on behalf of myself and my co-authors.